# A theoretical analysis of complex armed conflicts

**Sergio Rinaldi[1,2], Alessandra Gragnani[2], Francesco Niccolò Moro[3]\*, Fabio Della Rossa[2]**

**1** International Institute for Applied Systems Analysis (IIASA), Laxenburg, Austria, **2** Dipartimento di Elettronica, Informazione e Bioingegneria, Politecnico di Milano, Milano, Italy, **3** Dipartimento di Scienze Politiche e Sociali, Università di Bologna, Bologna, Italy

\* francesco.moro4@unibo.it

## Abstract

The introduction and analysis of a simple idealized model enables basic insights into how military characteristics and recruitment strategies affect the dynamics of armed conflicts, even in the complex case of three or more fighting groups. In particular, the model shows when never ending wars (stalemates) are possible and how initial conditions and interventions influence a conflict's fate. The analysis points out that defensive recruitment policies aimed at compensating for suffered losses lead to conflicts with simple dynamics, while attack groups sensitive to the damages they inflict onto their enemies can give rise to conflicts with turbulent behaviours. Since non-governmental groups often follow attack strategies, the conclusion is that the evolution of conflicts involving groups of that kind can be expected to be difficult to forecast.

**Data Availability Statement:** All relevant data are within the manuscript.

**Funding:** The authors received no specific funding for this work.

## Introduction

The interest in mathematical models of armed conflicts has increased in the last decades [1–4] owing to the wide prevalence and complex nature of such conflicts. Armed conflicts often involve more than two groups, which typically differ in their military characteristics and recruitment policies [5]. Thus, traditional descriptive conceptual models, like those suggested long ago by Lanchester [6] and Richardson [7–9] (for a review, see [10]), are of limited help.

The idealized model we propose indicates that stalemates, i.e., conflicts with no winner, arise. Stalemates can be stationary or have recurrent ups and downs in the army sizes and in the inflicted and suffered losses. Second, the model shows how outcomes depend on initial conditions. Third, when a conflict is trapped in a non-stationary stalemate, the model suggests when a short but heavy military intervention should be performed in order to have the highest chances of eradicating the enemies.

The results can be summarized with two statements: "brute force is often rewarding" and "defensive strategies are stabilizing (they lead to constant violence dynamics) while attack strategies are destabilizing (they can lead to periods of violence alternated with periods of relative peace)". They are obtained under the assumption that groups follow either defense strategies (their recruitment process depends only on their losses) or attack strategies (their

**Competing interests:** The authors have declared that no competing interests exist.

recruitment depends only on the injuries they inflict to their enemies). But they are also shown to remain valid under more realistic assumptions, i.e., when groups follow mixed strategies.

It is generally believed that the fate of complex conflicts cannot be predicted because data are too scarce and unreliable [11, 12]. The results obtained with the model show that the truth may be even less convenient: as the model reveals, the dynamics among the fighting groups can be extremely sensitive to initial conditions and parameter values. This suggests that a conflict's fate may practically remain unknown even if very rich data sets are available. This also resonates with contemporary readings of one of the most famous military studies, which finds that a key contribution of the Prussian general Clausewitz has been to recognize the chaotic and often non-measurable nature of military conflicts and their outcomes [13].

The paper is organized as follows. In the next section, the model is presented for a generic conflict involving any number $N$ of fighting groups. Then, in the following section a detailed analysis of the model is carried out in the case $N = 2$ under the assumption that groups follow either defense or attack strategies. In particular, it is proved that conflicts between defense groups can only have stationary stalemates, while conflicts involving at least one attack group can have periodic stalemates. Then, the analysis is extended to conflicts between three or more groups involving attack groups and the main result is that in these conflicts stalemates can also be aperiodic. Then, numerical bifurcation analysis is used to identify the influence that military and recruitment traits have on the fate of the conflict. Particular attention is devoted to the problem of eradication of the enemies either through long lasting structural interventions or through short but heavy military attacks (shocks).

Many of the obtained results are in agreement with common wisdom and can be recognized in past or contemporary conflicts, some of which are briefly discussed throughout the paper. Some other results, in particular those concerning shock interventions, are more difficult to parse. Finally, the main conclusions obtained with the idealized model are shown to remain valid in more realistic contexts, for example, when conflicts are influenced by unknown exogenous factors, when groups follow mixed strategies, when some groups are not easily detectable because hidden in civil populations, or when they are influenced by the success of other groups involved in other conflicts.

The potential readers of this paper comprise scientists working in applied mathematics or systems analysis as well as peace science and conflict resolution. To make the paper accessible to everyone, we intentionally avoid engaging in technical social science jargon and relegate all mathematical details to the Appendix in S1 File.

Further research is needed to address important open questions. Some of these are more academic in nature: Can the peaks of the losses exhibit simple dynamics? Can the patterns discovered in the frequency and severity of violent attacks through huge statistical analyses [14–16] (be a priori expected on the basis of the model? Which is the role of spatial dynamics and noise on conflict dynamics? Other questions are also of great relevance to decision makers: Can the model describe the effects of temporary coalitions or of temporary supports given to the enemies of the enemies? What is the impact of different choices of troops deployment over different conflict areas?

## The idealized model

In this section we present an idealized model of armed conflicts involving a finite number of groups $i = 1, \ldots, N$ fighting one against the other. To keep the complexity of the model under control, we assume that each group is described, at any time $t$, by a real non-negative variable $x_i(t)$, called *size* of the group (measured in suitable units) [17, 18]. Thus, the time derivative $\dot{x}_i$

of the size of the group is simply the unbalance between all regeneration and consumption flows.

If a group has no enemies its regeneration flows are the *basic recruitments* of soldiers, hired professionals, and new armaments, while the consumption flows are natural death, retirement, and obsolescence. When the group is small, all these flows can reasonably be assumed to be linearly related with the size of the group, so that

$$\dot{x}_i = b_i x_i$$

where $b_i$ is positive if the basic recruitment overcomes the sum of the consumption flows as it usually does in governmental groups. But if the group is large, various problems, like difficulties in the progress of careers or limitations in salaries and benefits, give rise to a surplus of retirement which, therefore, increases more than linearly (e.g., quadratically) with $x_i$. Thus, in the absence of enemies, the size of a group varies in accordance with a standard logistic equation

$$\dot{x}_i = b_i x_i - c_i x_i^2.$$

In general, in governmental groups, $b_i > 0$, so that $x_i$ tends asymptotically toward the *characteristic size $b_i/c_i$*. But in non-governmental groups (e.g., rebels, terrorist organizations, . . .) $b_i$ can be negative, because basic recruitment is subject to more constraints [19]. In these cases, $\dot{x}_i < 0$, i.e., in the absence of enemies, these groups go gradually extinct.

If the group is fighting against some enemies, two extra consumption and regeneration flows must be considered. The consumption flow, indicated with $L_{+i}$, is the total losses suffered by the group. The regeneration flow, indicated with $R_i$, is an extra recruitment of a mix of soldiers, other type of fighters (such as militias) and armaments: In an abstract sense, $R_i$ represents the overall *reaction* of the government and of the people supporting the group to the current state of the conflict, i.e., to the total losses $L_{+i}$ suffered by the group and to the total losses $L_{i+}$ inflicted to the enemies.

Thus, in conclusion, each group is described by the differential equation

$$\dot{x}_i = b_i x_i - c_i x_i^2 + R_i - L_{+i} \tag{1}$$

where the dependence of the reaction $R_i$ on the losses $L_{+i}$ and $L_{i+}$ and the dependence of these losses on the sizes of the groups must be specified.

For simplicity, the reaction $R_i$ is assumed to depend on a single variable $z_i$, i.e., $R_i = R_i(z_i)$, where $z_i$ is a weighted sum of the suffered and inflicted losses

$$z_i = \lambda_i L_{+i} + (1 - \lambda_i) L_{i+} \quad 0 \le \lambda_i \le 1$$

For $\lambda_i$ close to one the recruitment is aimed at compensating the losses suffered by the group, thus revealing a preference for defense strategies, while if $\lambda_i$ is close to zero the recruitment depends mainly on the losses inflicted to the enemies (attack strategies). For this reason, $\lambda_i$ is called *propensity to defend*. The function $R_i(z_i)$ must satisfy a few obvious properties. First, it vanishes at the origin ($R_i(0) = 0$) because in the absence of injuries an extra-recruitment is unjustified. Second, it is increasing (more injuries stimulate greater reactions) but bounded ($R_i(z_i) < R_i^{max}$) because the recruitment rate is limited by physical and economic constraints.

As for the losses, let us denote with $L_{ij}$ the loss inflicted by group $i$ (the killer) to group $j$ (the victim). Hence,

$$L_{i+} = \sum_j L_{ij} \quad L_{+i} = \sum_j L_{ji}$$

where the two sums are extended to all groups $j$ fighting against group $i$. The loss $L_{ij}$ depends on $x_i$ and $x_j$, but also on the sizes $x_l$ of all other groups fighting against group $i$, and must satisfy a few obvious properties. First, it vanishes when there are no killers ($x_i = 0$) or no enemies ($x_j = 0$). Second, it increases with respect to $x_i$ (more killers imply more victims) and, for simplicity, this dependence is here assumed to be linear, so that we can write

$$L_{ij} = x_i l_{ij}$$

where $l_{ij}$ (independent on $x_i$) is the damage produced in group $j$ by each unit of group $i$. The loss $L_{ij}$ increases also with $x_j$ (more enemies imply more damages) but is bounded, because the damage $l_{ij}$ is limited since the time spent in a conflict can not be entirely devoted to kill and destroy enemies (for example, prisoners require time to be handled and generate a lot of related activities here called bureaucracy). Finally, $L_{ij}$ (and, hence, $l_{ij}$) is decreasing with $x_l$, $l \neq j$, because each unit of group $i$ is forced to devote time to all its enemies, thus reducing the pressure on each particular group $j$.

Under these assumptions on $R_i$ and $L_{+i}$, model (1) is positive (i.e., the sizes of the groups can not become negative) and bounded (i.e., the sizes of the groups can not grow indefinitely). The first property follows from the fact that for $x_i = 0$ the recruitment $R_i$ and the losses $L_{+i}$ vanish so that model (1) gives $\dot{x}_i = 0$, i.e., $x_i$ can not become negative. In contrast, the second property follows from the boundedness of recruitment and losses which implies that for large values of $x_i$ the quadratic term $c_i x_i^2$ is dominant in Eq (1), so that $\dot{x}_i < 0$, i.e., the size of group $i$ must decrease. This means that model (1) can not have unrealistic behaviors like those that some of the best known classical models of armed conflicts have (for example, Richardson's models [7–9]) are unbounded, while in models by Lanchester [6] and Deitchman [20]) the sizes of the groups can become negative—for a comprehensive view of the different models, see [10]).

In order to stress the differences between various recruitment strategies the groups involved in the conflict are assumed to be either defense groups (indicated as $D$ groups) or attack groups (indicated as $A$ groups). By definition, $D$ groups are characterized by $b_i > 0$ and $z_i = L_{+i}$, while $A$ groups have $b_i < 0$ and $z_i = L_{i+}$. Obviously, $D$ groups represent a rough approximation of many governmental groups, while $A$ groups mimic many non-governmental groups (including groups of rebels or terrorist organizations). In all cases, the reaction recruitment $R_i(z_i)$ is assumed to be given by

$$R_i(z_i) = \rho_i R_i^{max} z_i / (\rho_i z_i + R_i^{max}) \tag{2}$$

where the meaning of the two parameters $\rho_i$ and $R_i^{max}$ is pointed out in Fig 1A. The parameter

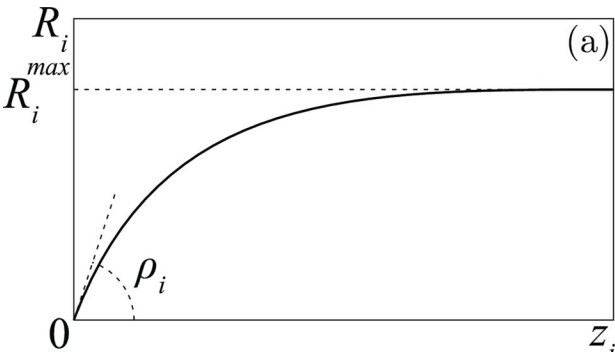
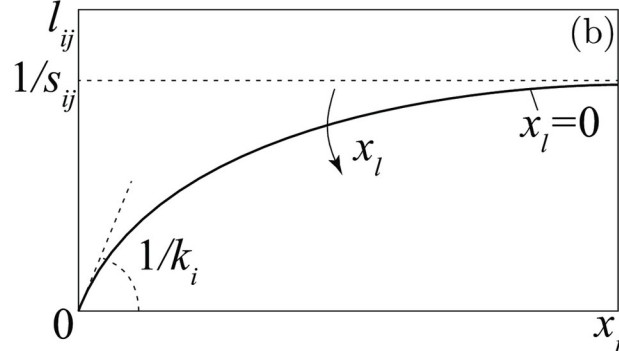

**Fig 1. The two functions involved in the model.** (a) recruitment policy (see Eq (2)); (b) military characteristic (see Eq (3)).

$\rho_i$, called *reactiveness*, is non-dimensional and represents the ratio between recruitment and loss when the loss is small. For this reason, we assume that $\rho_i < 1$ in $D$ groups, in order to exclude the unrealistic possibility that such groups recruit instantaneously more than the damages they suffer. As for the losses $L_{ij} = x_i\, l_{ij}$, we describe their dependence on the sizes $x_j$ and $x_l$ with the function

$$l_{ij} = x_j / \left( k_i + s_{ij} x_j + \sum_{l \neq j} s_{il} x_l \right) \tag{3}$$

which satisfies, as shown in Fig 1B, all the properties we have required. In Appendix 1 in S1 File we show that the parameter $k_i$, called *killing time* (its inverse $1/k_i$ is called *killing rate*), can be interpreted as the time spent by each unit of group $i$ to discover and attack one unit of enemies, while the parameter $s_{ij}$, called *service time*, is the time needed by each unit of group $i$ to take care of one unit of destroyed/captured unit of group $j$. Since the time $s_{ij}$ increases with the bureaucratic activities in which group $i$ is involved, we can conclude that bureaucracy limits the injuries that a group can inflict to its enemies, because

$$l_{ij}^{max} = \lim_{x_j \to \infty} l_{ij} = 1/s_{ij}.$$

In other words, more bureaucracy implies lower military effectiveness. In the last part of the paper, Eqs (2) and (3) are modified to deal with phenomena which are not taken into account in the idealized model.

Before entering in the discussion of the different types of conflicts, it is worth to note that our model hardly describes a real conflict as a whole. In fact, during a real conflict, various events not described by the model can suddenly perturb the sizes of the groups (e.g., an epidemic) or the military/recruitment characteristics (e.g., a radical technological change). These events, called perturbations, can be mathematically described as a sudden variation of the sizes of the groups or of its parameter values. Thus, a real conflict can be described by the model applied to each of the different segments identified by successive perturbations.

## Conflicts between two groups

The analytical treatment of model (1) is not possible when the number of groups is large. In contrast, conflicts involving only two fighting groups ($N = 2$) can be fully understood by combining the theory of two-dimensional dynamical systems (see, e.g., [21]) with numerical bifurcation analysis [22]. For this reason, this section is limited to the analysis of these conflicts, in the three possible cases shown in Fig 2, namely $D$–$D$, $D$–$A$, and $A$–$A$.

The results are presented graphically by showing time series of the sizes $x_1$ and $x_2$ of the two groups, or the corresponding trajectories in the two-dimensional space ($x_1$, $x_2$). Alternatively, we present time series and trajectories of the losses $L_{21}$ and $L_{12}$ suffered by each group. This is fully justified because the pair ($x_1$, $x_2$) can be proved to be in a one-to-one correspondence with the pair ($L_{21}$, $L_{12}$) (see Appendix 2 in S1 File). Numerical bifurcation analysis has been done using MATCONT: the interested reader can refer to [23–25], and references therein. The parameter values used to produce the figures are all listed in Appendix 3 in S1 File.

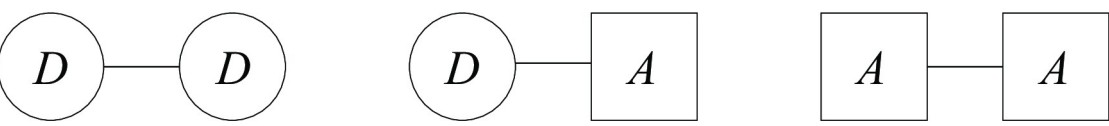

**Fig 2. The structure of the three possible conflicts between two groups ($D$ and $A$ are defense and attack groups, respectively).**

## *D—D* conflicts

*D–D* conflicts pit two groups with regular armies against each other. Fig 3 is a typical bifurcation diagram of model (1) in the case of a conflict between two defensive groups. In the diagram, the influence of two parameters, $b_1$ and $c_2$, on the dynamics is explored systematically. The top-left diagram reports the parameter plane $(b_1, c_2)$ (all other parameters are specified in Appendix 3 in S1 File), split into seven regions $i$ ($i = 1, \ldots, 7$), each one identifying a qualitatively different scenario. The curves delimiting the various regions are *bifurcation curves* (either *transcritical* bifurcation curves (*TC*) or *saddle-node* bifurcation curves (*SN*)). For any value of $b_1$ and $c_2$ in region $i$, the corresponding conflict has the time-evolution qualitatively sketched in panel $i$. Thus, for example, all conflicts in region 2 end with the victory of group 1 or with a stationary stalemate, depending on the initial sizes $(x_1(0), x_2(0))$ of the groups. Fig 3 says that *D–D* conflicts can have seven different long term outcomes, namely

- stationary stalemate (in region 1)

- stationary stalemate or 1 wins (in region 2)

- stationary stalemate or 2 wins (in region 3)

- 1 wins (in region 4)

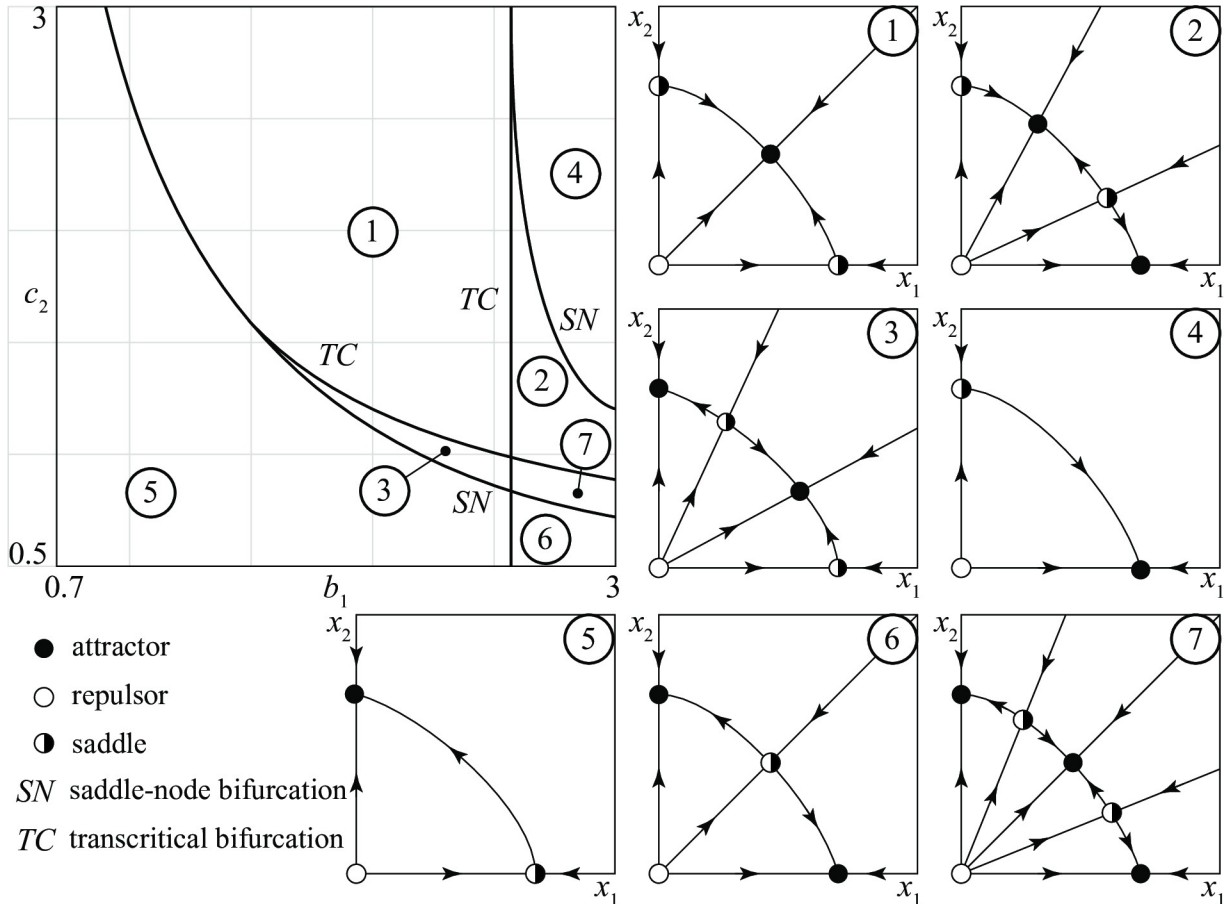

**Fig 3. Example of a bifurcation diagram of a *D—D* conflict.**

- 2 wins (in region 5)

- 1 wins or 2 wins (in region 6)

- stationary stalemate, or 1 wins, or 2 wins (in region 7)

This list points out that the initial sizes of the groups matter, since only in three out of seven cases the fate of the conflict is the same for all initial conditions. Fig 3 also says that periodic stalemates are not possible since there are no closed trajectories (cycles) in any of the 7 regions identified in the bifurcation diagram. This result, illustrated in Fig 3 for the parameter setting reported in Appendix 3 in S1 File, actually holds in general, as proved in Appendix 4 in S1 File.

Fig 3 also shows that for being able to predict a conflict outcome it is crucial to

(*i*) have a precise knowledge of the initial conditions and

(*ii*) have a precise knowledge of the parameter values.

Requirement (*i*) is pointed out in panels 2, 3, 6, 7, where the existence of alternative stable equilibria implies that small variations of the initial state of the conflict can have great consequences on its fate. For example, in panel 2 if the initial state $(x_1(0), x_2(0))$ is just above the stable manifold of the saddle (the union of the two trajectories that approach the saddle) the conflict tends to a stalemate while, if it is just below it, group 1 wins the competition.

Statistical evidence points out that several interstate conflicts, in the ancient past as well as in more recent times, have been trapped in a stalemate [26]. One of the paths to exit stalemates is an accidental or intentionally provoked perturbation. Accidental factors such as adverse weather have long been considered important perturbations. The Peloponnesian war lasted almost three decades before end came in sight. Before eventually defeating the Athenians in the Battle of Aegospotami (405 BC), Spartans had finally achieved a breakthrough against the then superior Athenian navy because the latter had been caught in adverse weather and decimated the year before [27]. The battle of Waterloo (1815) was also affected by meteorological conditions: heavy thunderstorms struck the terrain where French forces were about to attack their Anglo-Dutch enemies, making the terrain unsuitable for a decisive action based on cavalry and artillery [28]. Other types of perturbations are connected with intentional factors that–though relatively small–had a disproportionate impact on an operation. For instance, doctrinal innovation in the use of infantry decisively contributed to break a typical case of prolonged stalemate, trench warfare in WWI. Such innovations emerged on the Western front in the German Spring Offensive (March 1918). After a successful experiment on the Eastern front (Battle of Riga, September 1917) and in Italy (Battle of Caporetto, October-November 1917), German forces infiltrated assault troops ("Stormtroopers") through enemy lines with the support of artillery to neutralize defenses in very specific areas [29]. While that innovation did not allow to secure victory in war, it did nonetheless contribute to effectively overcome enemy's defensive positions that had been stable for almost 4 years.

Requirement (*ii*) is a consequence of the existence of *SN* bifurcations. In fact, suppose that a conflict is represented by a point in Fig 3 just below the *SN* bifurcation curve separating region 2 from region 4 and assume that such a conflict is at the stalemate indicated in panel 2. If the basic recruitment of group 1 is slightly increased, the parameter $b_1$ is also increased so that the point representing the new conflict moves in region 4, just above the *SN* bifurcation. But in region 4 group 2 goes extinct. Thus, in conclusion, a microscopic change in the recruitment policy has triggered a macroscopic transition from a stalemate to the victory of one group. This transition, as well as the *SN* bifurcation curve, is often called *catastrophic* (or simply *catastrophe*). Similarly, conflicts represented by points in regions 3 just above the other *SN*

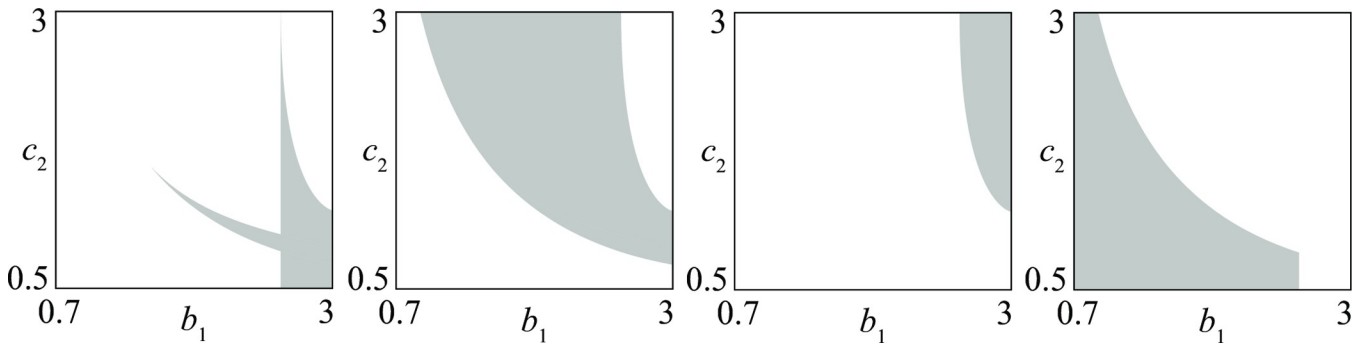

**Fig 4. Four different sets of $D$—$D$ conflicts (see text for their meanings).**

bifurcation, can have for a small change of $b_1$ a catastrophe starting from a stalemate and ending with the victory of group 2.

By suitably aggregating various regions $i$ of Fig 3, one can obtain subregions corresponding to conflicts of particular interest. For example, Fig 4 shows the following four sets of conflicts:

($a$) conflicts in which initial conditions matter (regions 2, 3, 6, 7 in Fig 3)

($b$) conflicts that can end in a stationary stalemate (regions 1, 2, 3, 7 in Fig 3)

($c$) conflicts in which the victory of 1 is guaranteed (region 4 in Fig 3)

($d$) conflicts in which the victory of 2 is guaranteed (region 5 in Fig 3)

The shape of these regions suggests conjectures on the influence that the parameters ($b_1$ and $c_2$ in the present case) have on the fate of the conflict. For example, Fig 4C shows that the conflicts in which the victory of 1 is guaranteed are in the NE corner of the bifurcation diagram where $b_1$ and $c_2$ are high. If we remember that the size of the army of group 1 when there is no conflict is $b_1/c_1$ (and similarly $b_2/c_2$ for group 2) we can conclude that the victory of 1 is guaranteed when the typical size of its army is relevant in comparison with that of group 2.

This result, consistent with common wisdom, is in line with the *brute force principle*: increase the size of the army to prevail in conflicts. This is compatible with insights of classic and modern military strategists that argue that superior numbers, and concentration of forces, have been the keys for victory at least until the advent of mechanized warfare. The French *levée en masse* of 1792 and the following use of nation-wide conscription have been often believed decisive in explaining the success of Napoleonic forces in the late XVIII and early XIX Century. In WWII, Germany constantly increased the pool of its conscripts, with the Wehrmacht resorting to drafting workers of the armament industry to sustain its effort in late 1944 [30]. The logic of "strength in numbers" was well alive in highly technological armies: General Westmoreland's continuous call for extra personnel in the Vietnam War was based on the assumption that victory was related to a given force ratio [31]. The brute force principle is also in line with the common practice of forming coalitions which has often proved to be rewarding. In manpower-intensive conflicts such as WWI, alliances have been clearly a tool to maximize the available manpower: Germany's decision to support Austria (leading to the outbreak of conflict) was partly due to the belief that the Austro-Hungarian army would have been essential to fight a war both on the Western and the Eastern fronts [32].

Once a first bifurcation diagram like that of Fig 4C has been produced, it is possible to obtain from it, through a numerical technique called *continuation* [22], similar bifurcation

diagrams with respect to other pairs of parameters. For example, in Fig 5 two new bifurcation diagrams (obtained from Fig 4C) point out conflicts where the victory of group 1 is guaranteed. The diagram in Fig 5A agrees with the brute force principle, showing that group 1 wins the competition if it increases its maximum recruitment rate $R_1^{max}$ or its killing rate ($1/k_1$).

Superior ability to mobilize recruits and material resources has been key in the Allies' ability to defeat Germany and Japan in WWII: the superior recruitment capacity, with full-scale intervention of the United States and the involvement of the Soviet Union, were combined with increased killing rates due to the higher US military industrial capacity [33]. Moreover, the cracking of the so-called Enigma code by British Intelligence gave advantage to the Allied side in the prolonged confrontation with German submarines [34], as it increased the Allies' killing rate by facilitating the discovery of German ships and consequently their destruction.

The interpretation of a bifurcation diagram is not always simple and straightforward, because the influence of a parameter can strongly depend on other parameters. For example, Fig 5B shows the case of a conflict where sufficiently high maximum recruitment $R_1^{max}$ guarantees the victory of group 1 only if group 2 has a sufficiently high service time $s_{21}$.

In the early stages of WWII, even before the German invasion (June 1941), the Soviet Union had radically increased the number of its soldiers resorting to mass mobilization, surpassing German forces in numbers [35] Nonetheless, the German army was able to penetrate Soviet defenses (end of 1941, beginning of 1942). A contributing factor that favored Germans–besides superior military technologies, doctrine, and command structures (Soviet commands had been purged by Stalin before and during the German campaign)–was the extreme brutality of German troops in dealing with the population inhabiting areas occupied with the advance (prisoners,potential collaborators, insurgents, and, more indiscriminately, Jews) [36], which reduced "service time" and contributed to a reduced bureaucratic burden that other approaches would have required.

### *D—A* conflicts

The main novelty with respect to the previous class of conflicts is that in *D–A* conflicts periodic stalemates are possible. Fig 6, where group 1 is a *D* group, shows one example of this kind of stalemate.

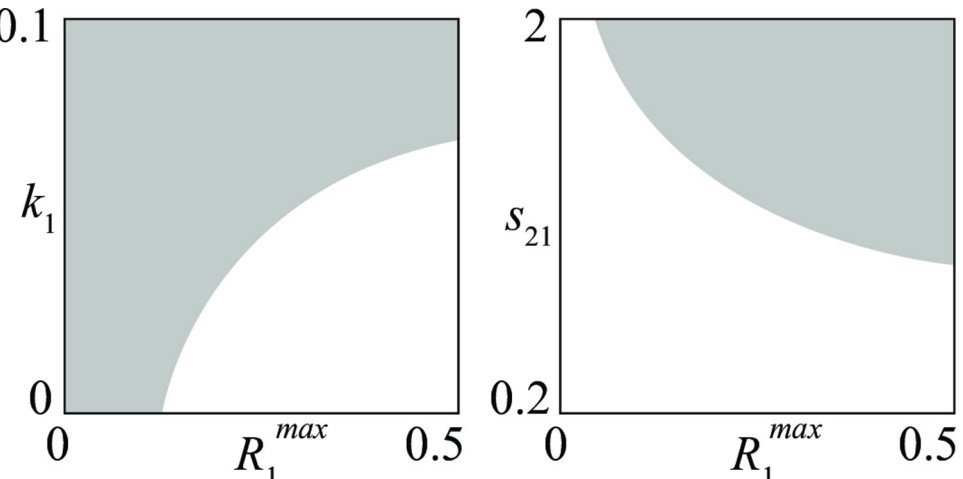

**Fig 5. Sets of *D—D* conflicts (shaded regions) where the victory of the first group is guaranteed.** (a) the brute force principle holds; (b) the brute force principle holds only if the second group has a heavy bureaucratic burden $s_{21}$.

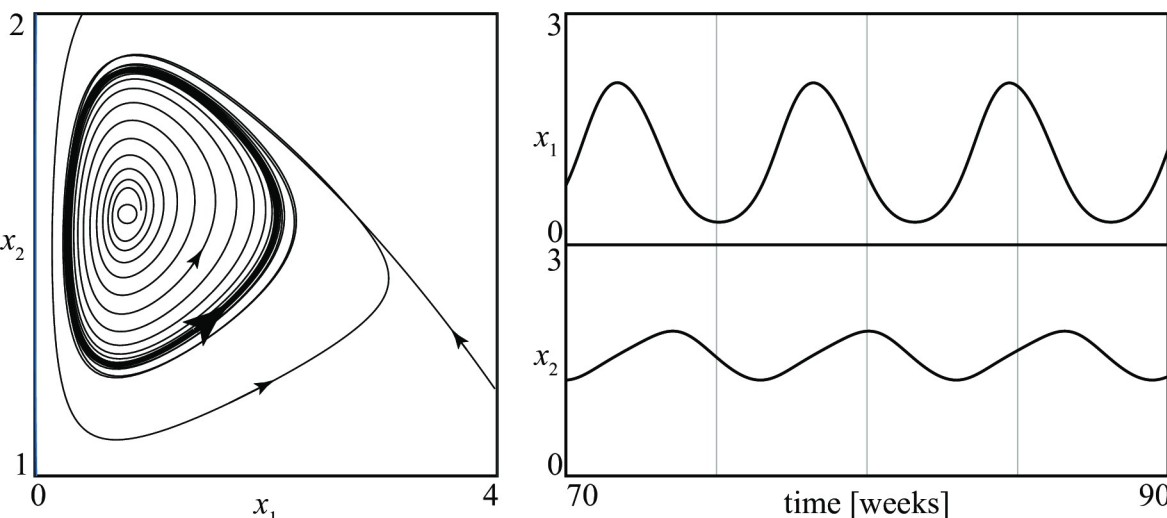

**Fig 6. An example of periodic stalemate in a *D—A* conflict.**

Bifurcation diagrams are more complex than in *D–D* conflicts because there are now two extra bifurcations (called *Hopf* (*H*) and *homoclinic* (*hom*) [21, 22]) that reveal the existence of cycles in the state portraits (periodic stalemates). A typical bifurcation diagram is shown in Fig 7 where the two free parameters are, as in Fig 3, $b_1$ and $c_2$. Thus, one should move to the right in this parameter space if the size ($b_1/c_1$) of the army of the *D* group in the absence of enemies is increased. Fig 7 shows that the bifurcation curves partition the parameter space into nine regions *i*, *i* = 1, . . ., 9, and that the possible long-term outcomes of the conflict are seven, namely

- *D* wins (in regions 1 and 9)

- one stationary stalemate (in regions 2 and 7)

- one stationary stalemate or *D* wins (in region 3)

- one periodic stalemate or *D* wins (in region 4)

- one periodic stalemate (in region 5)

- one stationary stalemate or one periodic stalemate (in region 6)

- two alternative stationary stalemates (in region 8)

The bifurcation diagram show that periodic stalemates exist in three regions (4, 5, 6) (the periodic stalemate shown in Fig 6 corresponds to a conflict in region 5). It also shows that there are no positive equilibria on the $x_2$ axis, i.e., the group following an attack strategy can never win the competition. From a conceptual point of view this is obvious, since in the idealized model *A* groups go extinct in the absence of enemies. However, the model shows that during the evolution of some conflicts the size of the *D* group temporarily can drop almost to zero, thus creating (in the real conflict) favorable conditions for a forceful removal of the leaders.

This is compatible with findings relative to revolutions (intended as attempts to overthrow a government) that would occur when the governments' forces have been defeated or depleted in wars: an example of this kind is the overthrow of the Tzarist regime in Russia (February

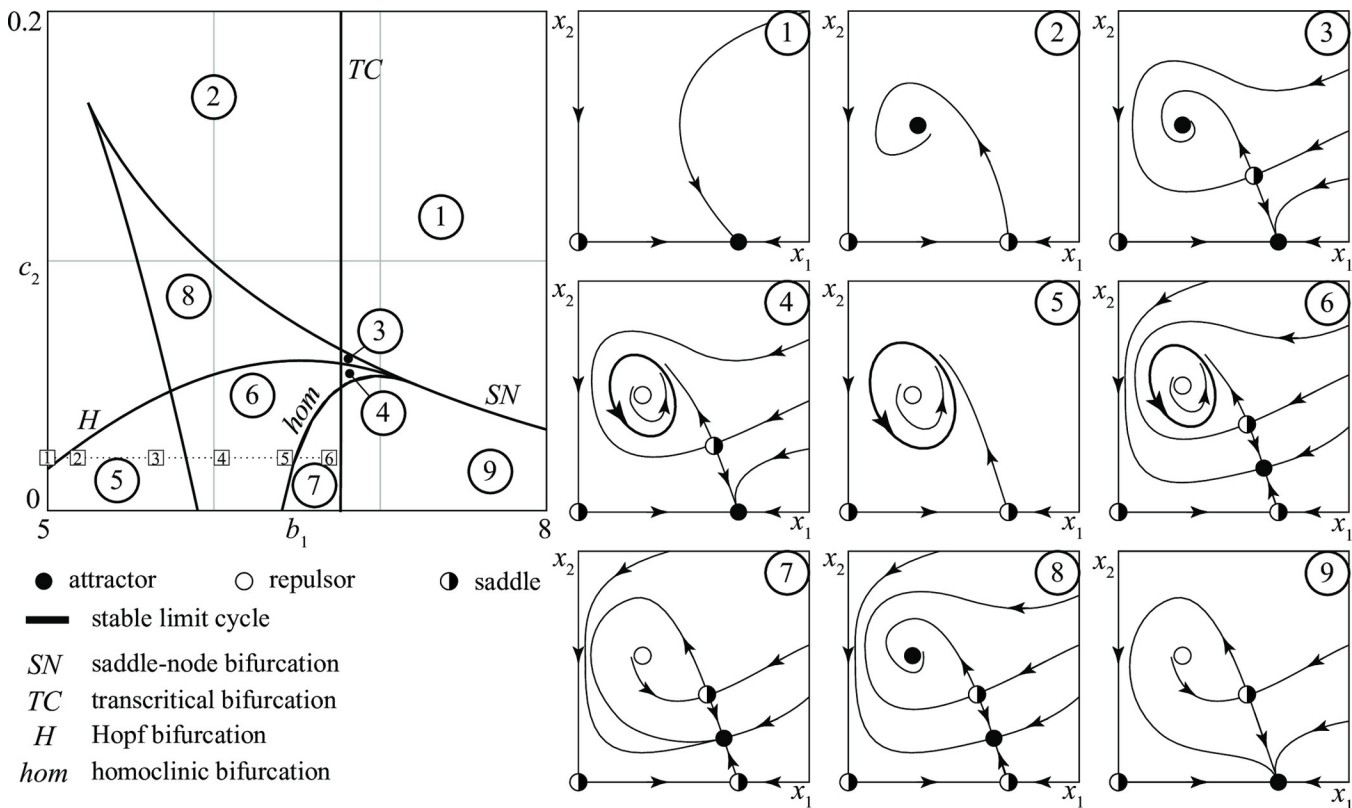

**Fig 7. Example of a bifurcation diagram of a $D$—$A$ conflict.**

1917) by a coalition of Republican and Socialist forces following the very harsh experience in WWI [37]. Many studies confirm that military depletion and defeat is not a necessary or sufficient cause for revolution, but positively affects its likelihood [38].

To be more precise on this issue, consider a conflict that starts with the $D$ group at its characteristic size $b_1/c_1$ and a few fighters following an attack strategy. Thus, the initial point of the conflict in the space $(x_1, x_2)$ is close to the equilibrium on the $x_1$ axis. Fig 7 shows that only in panels 2 and 5 the trajectory describing the evolution of the conflict passes close to the $x_2$ axis, where the overthrow of the government is facilitated. It is also interesting to notice that if point $(b_1, c_2)$ is in the SW corner of the central panel in Fig 7, the conflict is in region 2 or 5. In other words, if the basic recruitment ($b_1$) of the $D$ group is small and the $A$ group has a low resistance ($c_2$) to grow, there are high chances for a government overthrow.

As in the $D$–$D$ case, it is possible to extract from Fig 7 regions corresponding to particular conflicts like those reported in Fig 8, which have the following meanings

(*a*) conflicts in which $D$ can win (regions 1, 3, 4, 9 in Fig 7)

(*b*) conflicts in which the victory of $D$ is guaranteed (regions 1, 9 in Fig 7)

(*c*) conflicts with possible stalemates (regions 2–8 in Fig 7)

(*d*) conflicts in which only periodic stalemates are possible (region 5 in Fig 7)

The geometry of these regions suggests simple conjectures on the influence of the parameters. For example, Fig 8A points out that $D$ can eradicate $A$ provided its basic recruitment is

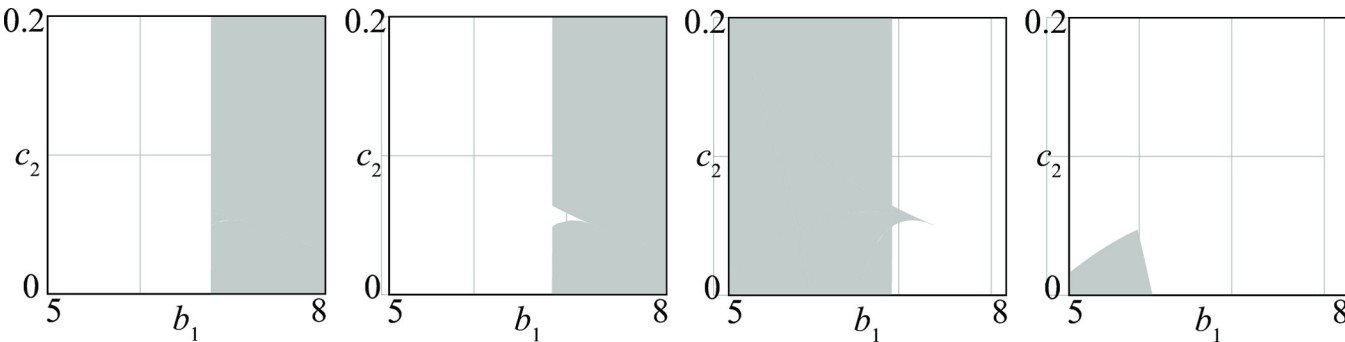

**Fig 8. Four different sets of $D$—$A$ conflicts (see text for their meanings).**

sufficiently large. Once more, this result is in agreement with the brute force principle. Similarly, Fig 8C shows that $A$ can remain in the game forever only if the size of the $D$ army is not too large.

Using the continuation method, bifurcation diagrams like those in Fig 8 can be transformed into similar bifurcation diagrams with respect to other pairs of parameters. Fig 9 shows two examples of bifurcation diagrams derived from Fig 8D where the shaded regions represent $D$–$A$ conflicts in which periodic stalemates are the only possible outcome. These diagrams point out that if the rate at which attack groups recruit is sufficiently high ($R_2^{max}$ and/or $\rho_2$ large) or if the time they need to find and attack an enemy is marginal ($k_2$ small), then periodic stalemates are likely to occur. This means that if groups following attack strategies become militarily more efficient, they have higher chances to avoid eradication by entering in a regime characterized by periodic ups and downs of their sizes and losses. The result we have obtained can be summarized in the statement: *attack strategies destabilize conflicts*.

Before closing this section, let us notice that conflicts with periodic stalemates correspond to a region in parameter space delimited by Hopf and homoclinic bifurcation curves (see curves $H$ and *hom* in Fig 7). Close to $H$ the periodic regimes are smooth and the differences between maximum and minimum losses decline when approaching the bifurcation. In contrast, close to *hom* the periodic stalemates are characterized by violent fights separated by periods of stasis that become longer when approaching the bifurcation. Thus, the periodic fights

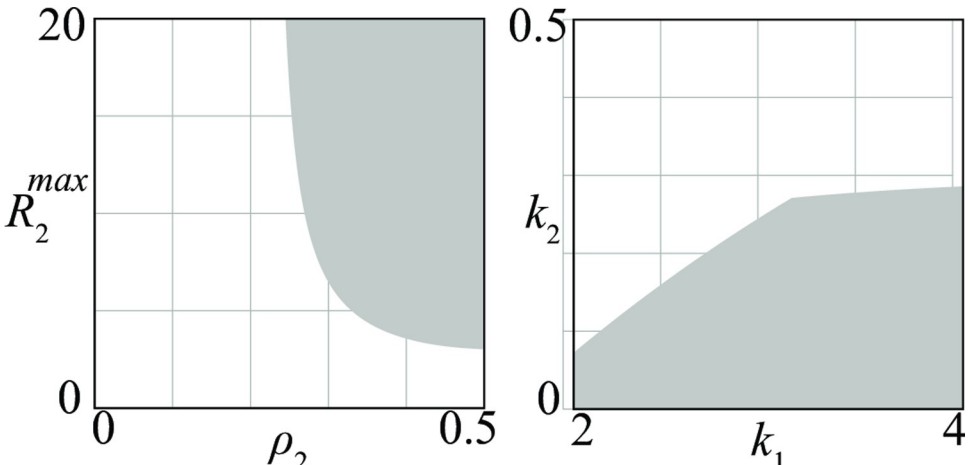

**Fig 9. Sets of $D$—$A$ conflicts (shaded regions) where a periodic stalemate is the only possible outcome.**

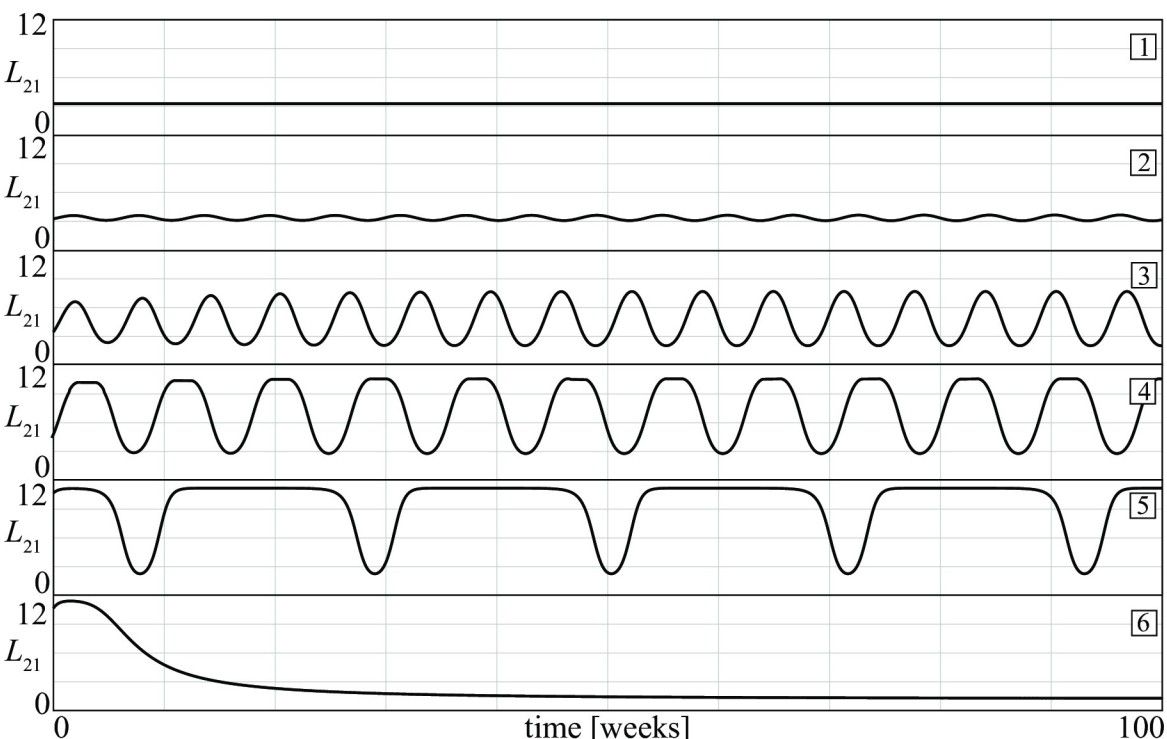

**Fig 10. Time patterns of losses $L_{21}$ in six different D—A conflicts (see points 1, . . ., 6 in Fig 7).**

can disappear either by becoming less and less violent (when $b_1$ is decreased) or by becoming less and less frequent (when $b_1$ is increased). Fig 10 shows the losses $L_{21}$ suffered by the $D$ group for the six different conflicts corresponding to points 1, 2, . . ., 6 in Fig 7. The figure shows that, starting from point 1, a continuous increase of the basic recruitment of the $D$ group first gives rise to small oscillations of the losses (point 2). Then, the oscillations become more relevant (point 3) and gradually quite violent but also more rare (points 4 and 5). Finally, when the homoclinic is crossed (between points 5 and 6) there is a catastrophic transition to a stationary stalemate characterized by lower injuries to the $D$ group. The mean loss $\bar{L}_{21}$ along the entire segment from point 1 to point 6 in Fig 7 first increases as shown in Fig 11 and then suddenly drops when the periodicity disappears.

Fig 7 shows that the aggressiveness of the attack groups destabilizes the conflict and also limits the validity of the brute force principle, which must therefore be followed with caution. Cases in point are several, Vietnam is a relatively straightforward example: even if the number

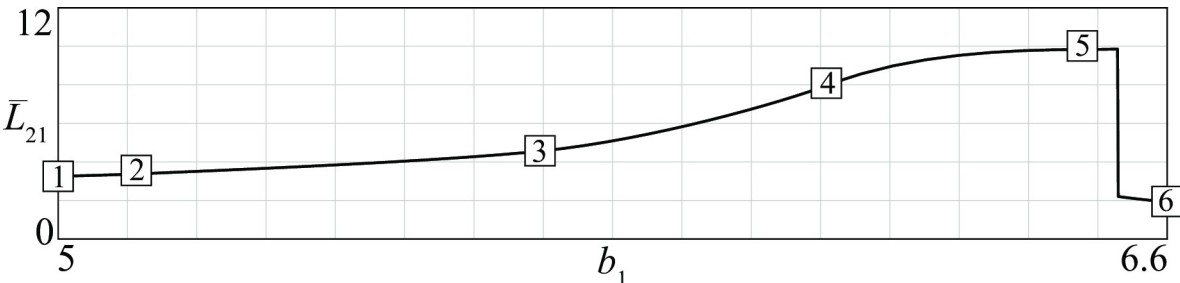

**Fig 11. Mean loss $\bar{L}_{21}$ when the basic recruitment of the D group is increased (see points 1, 2, . . ., 6 in Fig 7).**

of US troops largely increased in 1964–1968 period, leading to a very high number of losses of the North Vietnamese Army, the ability of North Vietnam's autocratic regime to keep fighting was not fundamentally reduced [39, 40]. Literature on wars of national liberation has observed how democracies fighting distant "small wars" often tended to lose such wars [41, 42]. Different interests at stake, different regime type and problems related to the ability of waging distant wars have explained US military performance across different wars in the past two centuries [43].

### *A—A* conflicts

We now present a few results concerning *A–A* conflicts, without entering into details because the aim of this paper is to focus on conflicts involving at least one defensive group.

The main characteristic of *A–A* conflicts is that extinction of both groups is not only possible but is actually guaranteed if the groups are initially small. In fact, when $x_i$ is small, model (1) is approximated by $\dot{x}_i = b_i x_i$ with $b_i < 0$ so that the sizes of the groups vanish at exponential rate. This means that fights between *A* groups are usually relatively short, because they both inexorably vanish if the event triggering the initial antagonism is not very severe.

But extinction is also guaranteed if the initial antagonism is large provided the groups are not very reactive ($\rho_i$ small). This is shown in Fig 12, where the boundary separating the coexistence and extinction regions is the concatenation of saddle-node (*SN*), homoclinic (*hom*), and heteroclinic (*het*) bifurcation curves. For this reason, in conflicts close to the *hom* or *het* curves, the groups evolve toward a periodic stalemate.

A complete bifurcation analysis would actually point out that the coexistence region is partitioned into 15 subregions (not shown) characterized by all possible combinations of one or two stationary or periodic stalemates (one stationary; one periodic; one stationary and one periodic; two stationary; two periodic). In general, when there are two stalemates one of them is dominated by group 1 (in the sense that $x_1$ is always greater than $x_2$), while the other is dominated by group 2, as shown in the right panel of Fig 12 which refers to a conflict with two alternative periodic stalemates.

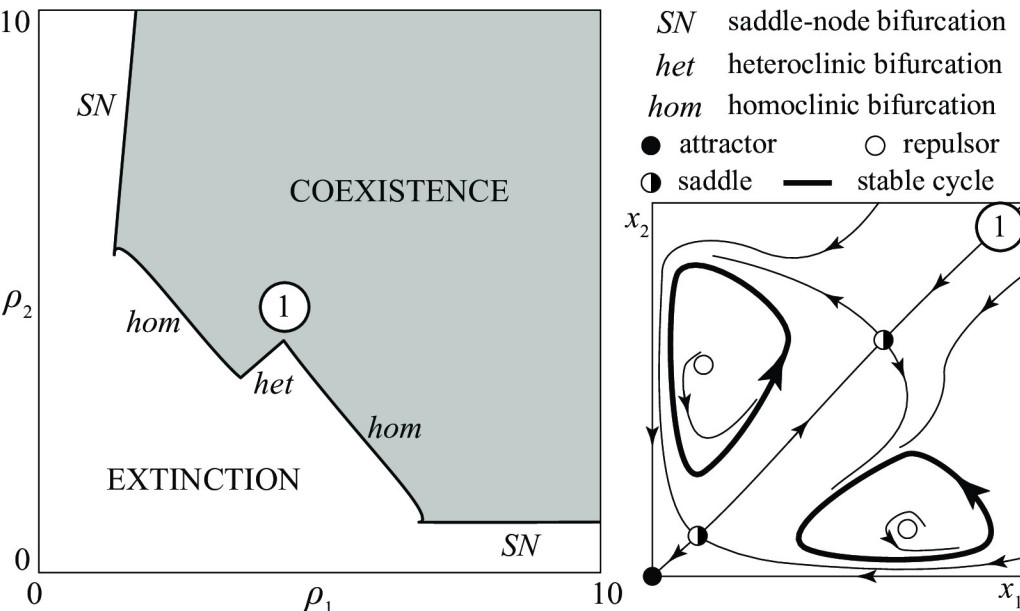

**Fig 12. Example of a bifurcation diagram of a *A—A* conflict.**

## Conflicts between three or more groups

In this section we show that the properties discovered so far continue to hold when the fighting groups are three or more. The validity of the brute force principle can be very easily checked through systematic simulations of the conflicts (not shown). In contrast, the fact that the addition of extra *A* groups destabilizes the conflict is more difficult to parse. However, it is not a great surprise since the addition of a third group in a conflict between two groups transforms the model into a three-dimensional nonlinear dynamical system for which complex behaviors are generically expected [21, 44].

While there are only three types of conflicts between two fighting groups (see Fig 2), in the case of three groups the conflicts can have ten different structures, as shown in Fig 13. Conflicts involving three or more actors are frequent. Contemporary conflicts in Iraq and Syria feature a plurality of actors, which are also heterogenous in terms of status, being both governmental and non-governmental groups ranging from quasi-states armies (such as the Kurdish Peshmerga) to terrorist groups (such as Al Qaeda in Iraq).

Bifurcation diagrams for conflicts involving three fighting groups are really complex. For example, the number of subregions in which the parameter space is partitioned by the bifurcation curves (7, 9, and 15 in the previous section) is not anymore bounded, because the bifurcation curves can be infinitely close one to each other. For this reason, we do not show bifurcation diagrams for these conflicts in this paper.

Of course, it would be interesting if something could be said about the existence of possible aperiodic stalemates (in one of the two classical forms (*quasi-periodic* or *chaotic*) predicted by the theory (see, e.g., [21, 22]) by looking only at the structure of the graphs describing the conflict. This is obviously a dream, because the structure of a system composed of various subsystems does not determine, in general, its properties. Many modeling experiences (see [45–47] for the very first contributions) carried out in population dynamics and neurosciences, as well as industrial automation and social sciences, suggest the use of a few rules of thumb based on simple topological indicators to capture the probability of existence of complex behaviors in systems composed of interconnected subsystems. One of these rules is that systems with many oscillating subsystems have high chances of complex behaviors. As shown in the previous

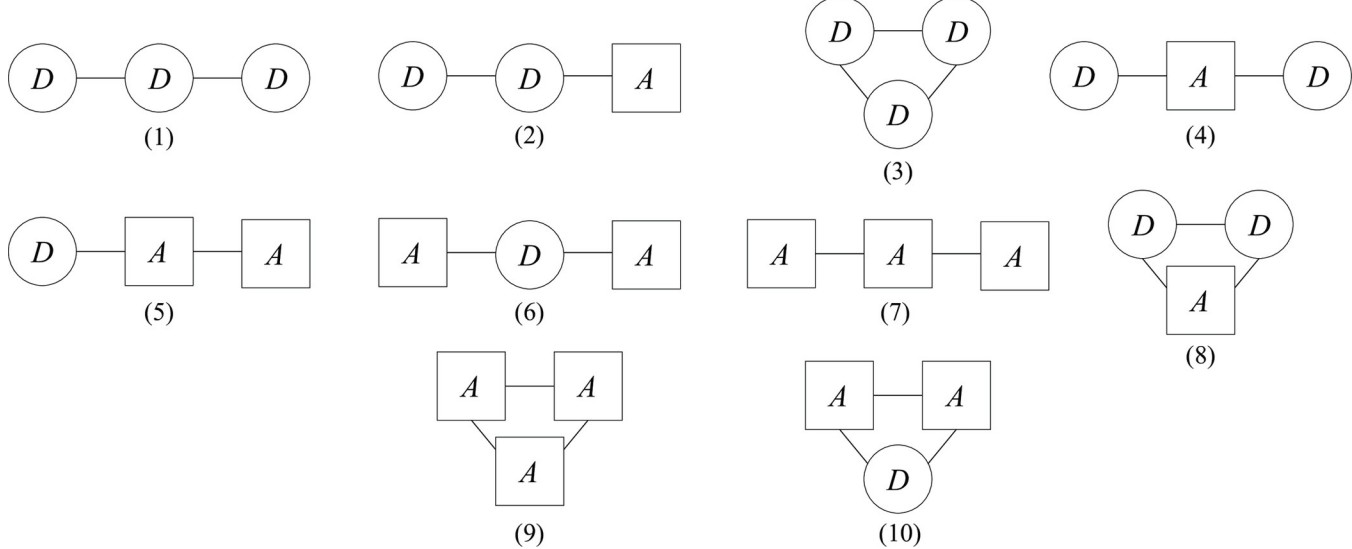

**Fig 13. The structure of all possible conflicts between three groups (*D* and *A* are defense and attack groups).**

section, in our case the elementary subsystems responsible of oscillatory behaviors (i.e., periodic stalemates) are the $D$–$A$ and $A$–$A$ conflicts between two groups. Thus, the rule we suggest is to evaluate the chances that aperiodic stalemates exist by simply counting the number $n$ of arcs of type $D$–$A$ and $A$–$A$ in the graph of the conflict. If we use this topological indicator in the conflicts described in Fig 13, we can immediately partition them into two groups

- conflicts (1), (2), (3) where $n = 0$ or $n = 1$

- conflicts (4), (5), . . ., (10) where $n = 2$ or $n = 3$.

Thus, our rule allows us to conjecture those conflicts of the first group can not have aperiodic stalemates, while conflicts of the second group can. This conclusion is in full agreement with our numerical experiments. Indeed, each conflict of Fig 13 has been simulated for 1000 randomly selected parameter settings and no aperiodic stalemate has been found for conflicts of type (1), (2), and (3), while for all other conflict types a few quasi-periodic and/or chaotic stalemates have been obtained. The form of the stalemate can be easily established from its Peak-to-Peak Plot (PPP) obtained by plotting pairs of subsequent peaks of a time series of the sizes or of the losses in a plane. In fact, if the PPP is a finite number of points, the stalemate is periodic, if it is a finite number of regular curves, the stalemate is quasi-periodic, while fractal PPP's are footprints of chaotic stalemates [48].

A first example of an aperiodic stalemate is shown in Fig 14 in the space of the losses $L_{+i}$. The figure refers to a conflict of type (5) in Fig 13. Actually, this is the first "military" chaotic stalemate derived from a model. Its PPP (see Fig 14B where $L_{+3}^*$ is the $k$-th peak of $L_{+3}$ and $L_{+3}^{**}$ is the $(k+1)$-th peak of $L_{+3}$) is a fractal set, confirming that the stalemate is chaotic. The corresponding time series of the losses in Fig 14C shows very clearly that the coexistence regime of the three groups is not periodic. In particular, the time series of the losses suffered by the defensive group points out that the severity of the injuries and their frequency of occurrence vary irregularly as observed in civil war dynamics (for a review, see [49]). The same figure also shows that the conflict enters, from time to time, in a period of fibrillation (see the time series of the losses), a phenomenon that has been observed in some cases.

Fibrillations, intended as rather frequent spikes in the levels of violence, can be easily found in multi-actor conflicts. The Iraq War (started with the American intervention of 2003), provides a snapshot of such phenomenon, being characterized by the presence of multiple governmental actors (the major being the new Iraqi government after the demise of the previous regime and the American government) and several non-state actors (to name just the most

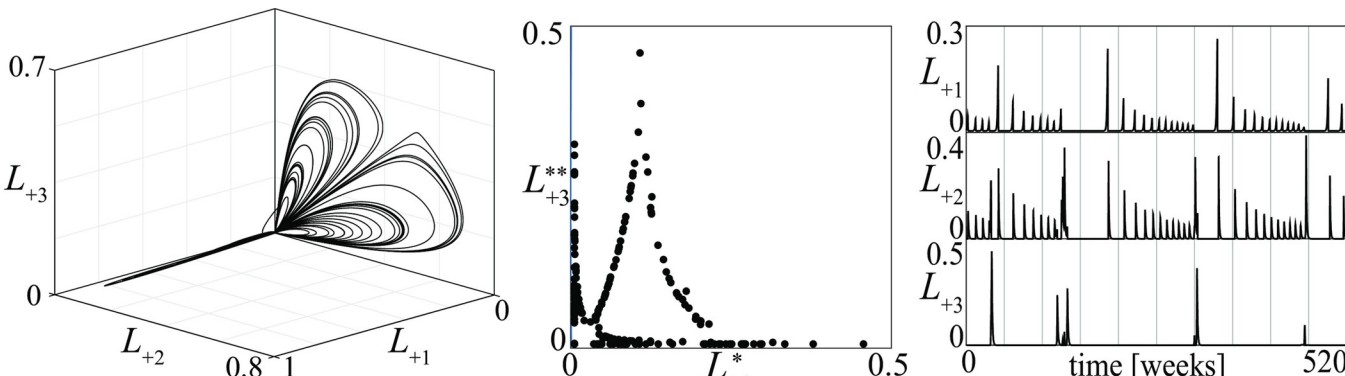

**Fig 14. A military chaotic stalemate for a conflict of type (5) in Fig 13.** (*a*) the stalemate in the three dimensional space of the losses $L_{+i}$; (*b*) the PPP where $L_{+3}^*$ and $L_{+3}^{**}$ are subsequent peaks of the losses of the third group; (*c*) ten years long segments of the corresponding time series.

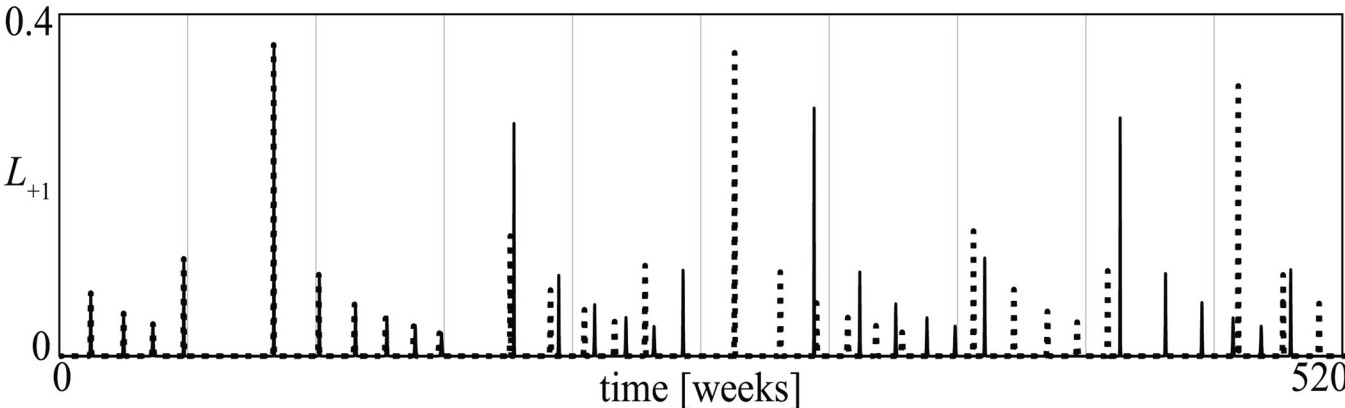

**Fig 15. Time evolution of the losses in the conflict described in Fig 14.** The continuous curve is as in Fig 14 while the dotted one has been obtained for a 1% variation of the initial loss of the *D* group.

important: Sunni insurgents, Shi'a militias in Baghdad and the South, Kurdish forces in the North). After a few years (2008–2012) in which violence had receded following an intensification of American presence (the so-called "surge" of 2007), the years 2013 and 2014 featured an intensification of violence, with highly irregular spikes in certain weeks (early June and early August 2014, for instance, see [50], year 2018).

The main property of chaotic stalemates–known as *sensitivity to initial conditions* [21, 51]–is illustrated in Fig 15 for the conflict described in Fig 14. The two time series of the losses of the *D* group are initially indistinguishable because the initial conditions differ only 1%, but then strongly diverge one from the other at a rate dictated by the so-called maximal Lyapunov exponent [21, 52]. Fig 15 shows that the time evolution of a conflict in a chaotic stalemate is *unpredictable* even if the initial conditions of all groups are almost precisely known.

Once a chaotic stalemate has been obtained, it is possible to detect, through simulation, the impact that any parameter change has on its characteristics. For example, varying the reactiveness of the *D* group from $\rho_1 = 0.05$ to $\rho_1 = 0.5$, the attractor and the associated time series of Fig 14 become like in Fig 16. A comparison of the two figures shows that the reactiveness $\rho_1$ has a quite relevant impact on the amplitude and frequency of the losses.

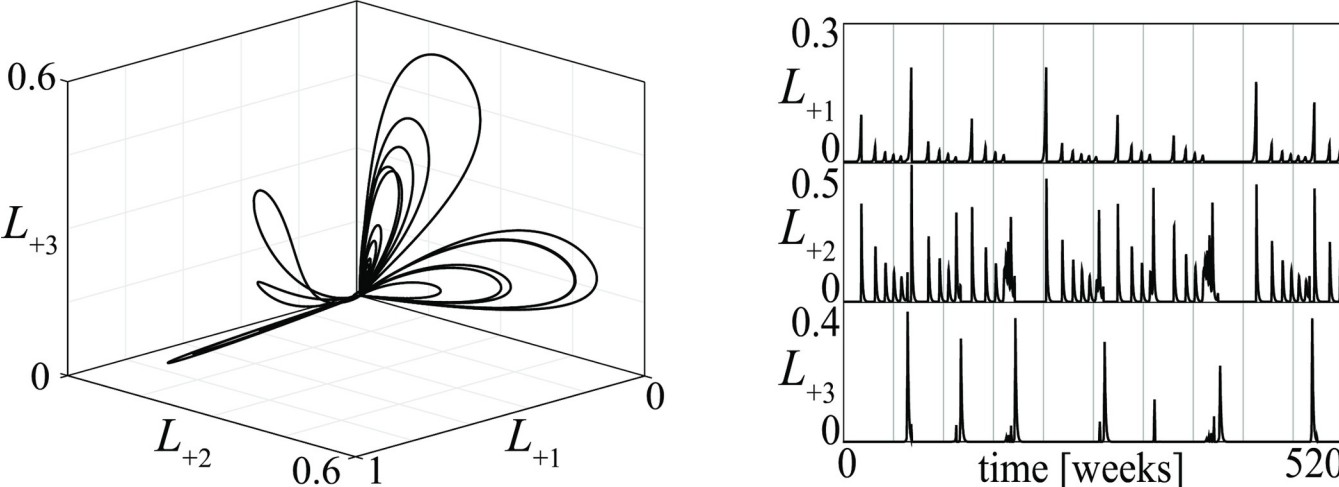

**Fig 16. Stalemate of the same conflict described in Fig 14 but with an increased reactiveness $\rho_1$ of the D group (from $\rho_1 = 0.05$ to $\rho_1 = 0.5$).**

Two other aperiodic stalemates are shown in Fig 17. The first is quasi-periodic (its PPP is composed of regular curves) and refers to a conflict of type (6) in Fig 13, while the second is chaotic (its PPP is a fractal set) and refers to a conflict of type (9) in the same figure. Figs 14, 16 and 17 show that aperiodic stalemates can be very different in different conflicts.

The topological arguments we have used for conflicts between three groups are obviously general. Thus, we should expect strong turbulences even when there are more than three groups provided the sources of instability are many. For example, consider the conflict between five groups described in Fig 18A: two independent *D* groups, each one fighting against two *A* groups, one being in common. In this conflict there are four sources of potential instability because all arcs in the graph are of type *D*–*A*. This case can again be illustrated by the fight against the so-called Islamic State (IS), which intensified after 2014. The governments of Syria and Iraq both fought against IS, while simultaneously facing different hotspots of insurrections at home [53]. Thus, our rule applied to this conflict predicts that very complex chaotic stalemates can exist. This is indeed the case, as illustrated in Fig 18B, where an unbelievably complex stalemate is reported by showing the time patterns of all losses for a five-year period.

The relation between the chaoticity of stalemates and the presence of multiple *A* groups is coherent with literature focusing on the effect of the fragmentation of agency in intrastate wars. In fact, presence of multiple rebel groups has been found to increase the intensity of

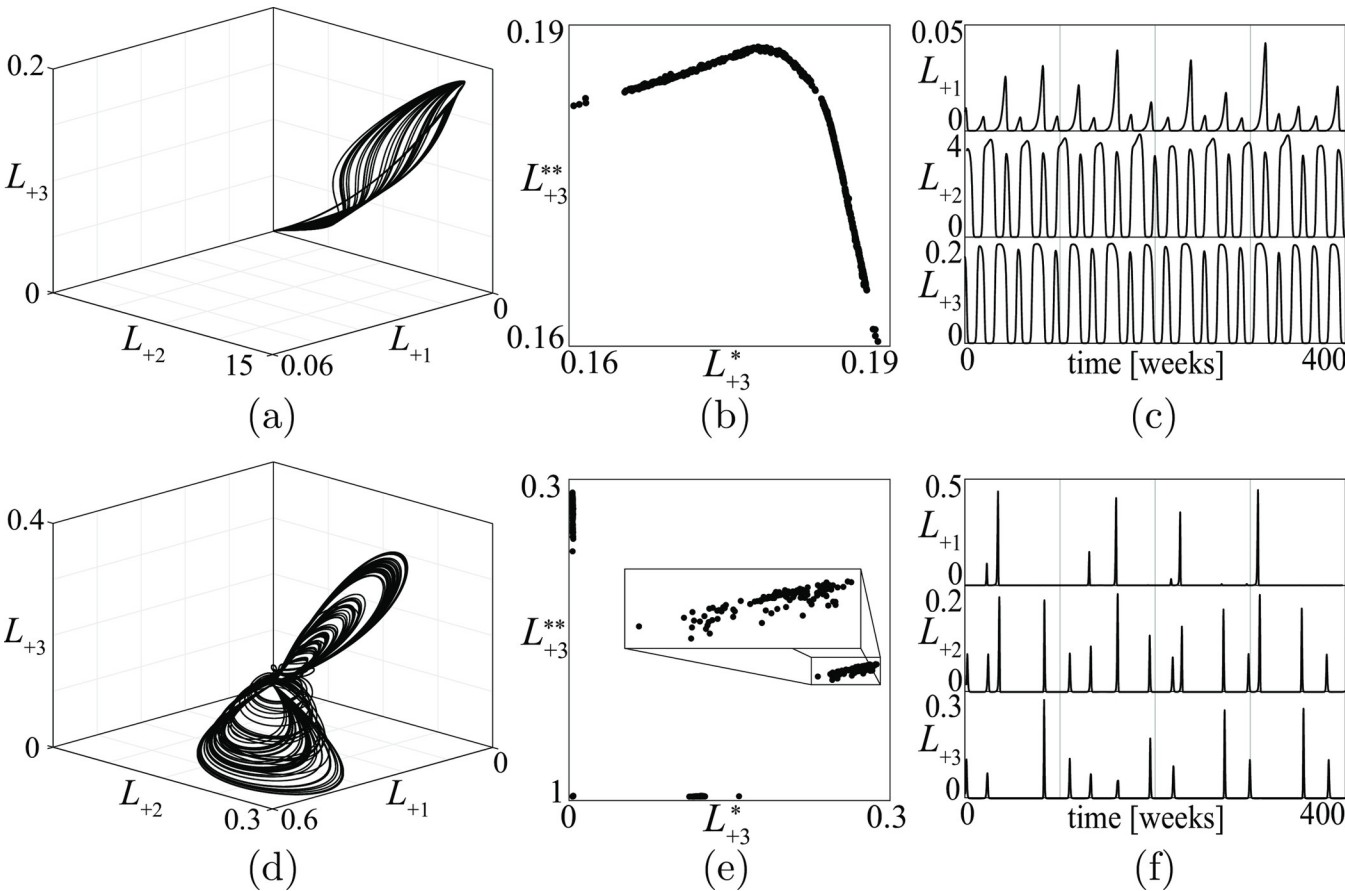

**Fig 17. Aperiodic stalemates in conflicts between three groups.** Top: quasi-periodic stalemate for a conflict of type (6) in Fig 13. Bottom: chaotic stalemate for a conflict of type (9) in Fig 13.

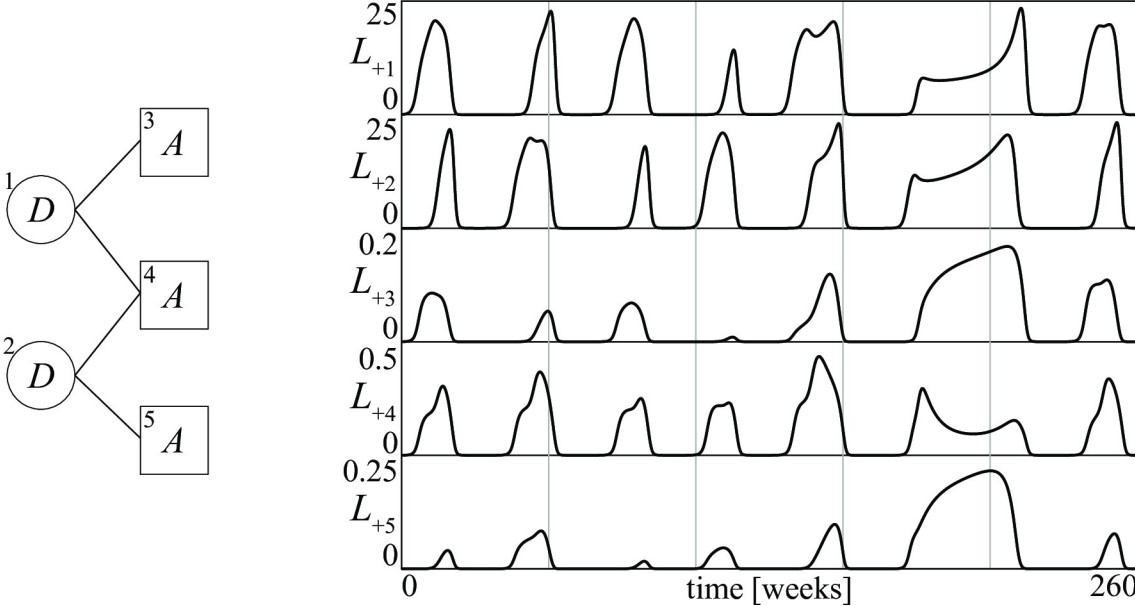

**Fig 18.** A conflict between five groups (*a*), and the time patterns of the losses of its chaotic stalemate (*b*).

attacks against the state [54], to give rise to internecine war among insurgents [55], to prolong hostilities and diminish the chances of reaching settlements [56] and durable peace [57].

## Eradication of the enemies

When a conflict is trapped in a stalemate the problem of major concern of any group is the eradication of the enemies (i.e., what should be done to interrupt the stalemate and win the competition). Historical conflicts show that groups have followed two different ideas to achieve breakthroughs in stalemates.

The first is to realize a structural change that modifies a strategic military or recruitment characteristic of the group. Very often the choice of the characteristic to be modified is suggested by the brute force principle. An example is WWII and the development of new technologies during the war. This idea is clearly illustrated by the bifurcation diagram of Fig 5A: an increase of $R_1^{max}$ or $1/k_1$ triggers a switch from a stalemate to the guaranteed victory of the first group.

The second idea is to try to eliminate the enemies forever by reducing their size at a favorable moment through a shock, i.e., through a short but heavy military intervention (often performed with the help of an allied country not directly involved in the conflict). The protracted war in Sri Lanka, pitting the government against the so-called Tamil Tigers (LTTE, Liberation Tigers of Tamil Ealam), has been ended by after 25 years by an intensification of the military activities of the government leading a crucial 15-month offensive that decimated enemy forces [58]. "Shock and awe" attempts are far from being always successful, though. Air campaigns such as Rolling Thunder, conducted by the United States against North Vietnamese forces since March 1965 to coerce North Vietnam to stop support of irregular forces and direct intervention in the South Vietnam, proved to be largely ineffective. Designed to last eight weeks, it actually lasted for more than three years with no effect on the overall outcome of hostilities [59]. This idea can be illustrated by a state portrait: a shock reduces instantaneously the size of the enemy so that the state of the conflict after the shock can be in the basin of attraction of the

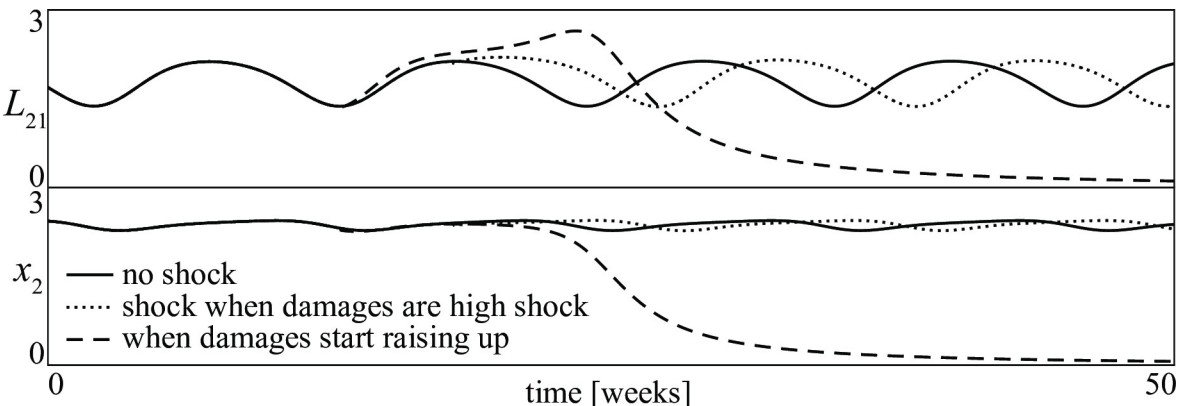

**Fig 19. Losses $L_{21}$ of the D group and size $x_2$ of the A group in a D—A conflict.** (*continuous curves*) without any intervention the conflict tends toward a periodic stalemate; (*dotted curves*) with a shock given when the damages are high the *A* group is not eradicated; (*dashed curves*) with a shock given when the damages start raising up the *A* group is eradicated.

equilibrium corresponding to the victory of one of the groups. This happens for example when a *D–A* conflict is in the stationary stalemate of region 3 in Fig 7. The state of the system after a heavy shock can be below the stable manifold of the saddle, i.e., in the basin of attraction of the equilibrium on the $x_1$ axis.

When a conflict is in a non-stationary stalemate a relevant problem is to choose the time at which the shock should be performed to be successful. Similar problems arise in agriculture (pest control), forest management (fire prevention), epidemics (vaccination), medicine (fibrillation), etc. In many of these cases, experience has proved that shocks given when the enemies are strong are often unsuccessful. In contrast, shocks given when the enemies are not yet a problem can be more rewarding (for example, campaigns against mosquitos should be started very early, when their damages are not yet relevant). In conclusion, the suggestion is: attack the enemies as soon as damages are perceived to increase. Our model supports this suggestion. This is shown in Fig 19, where the losses of the first group and the size of the second group are shown for three different assumptions: no shock (continuous curve); shock when damages are high (dotted curve); shock when damages start raising up (broken curve).

## Robustness of the results

The results obtained so far have been derived from an idealized model that describes an armed conflict under extremely simplified assumptions. We now show that these results, in particular the fact that defense [attack] strategies are stabilizing [destabilizing], continue to hold under more realistic assumptions. We limit our analysis to four aspects that are neglected in the idealized model, and we study, mainly through simulation, the impact they have on the conflict. To obtain easily interpretable results, we study the model by introducing only one neglected aspect at a time.

## Exogenous factors

We start by removing the main simplifying assumption made so far, namely the absence of exogenous events (political, social, physical, . . .) influencing the conflict. For this, we modify the structure of the model assuming that an *exogenous stress* $w(t)$ acts on the conflict, as shown in Fig 20. Thus, under quite general assumptions, the conflict can be described by two sets of ordinary differential equations.

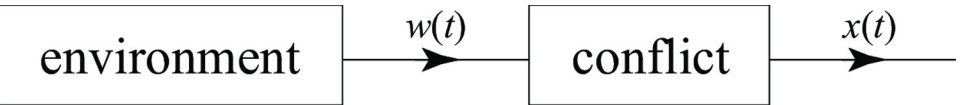

**Fig 20. The modified model when an exogenous stress $w(t)$ is influencing the conflict.**

$$\dot{w}(t) = g(w(t)) \qquad \dot{x}(t) = f(x(t), w(t))$$

The first $M$ equations ($\dot{w} = g$) describe the exogenous stress and the remaining $N$ equations ($\dot{x} = f$) describe the conflict between $N$ groups. The second set of equations is the one we have used until now where, however, the parameters are not constant anymore because at least some of them are influenced by the exogenous stress $w(t)$. The question that naturally arises is "who is responsible of chaoticity of the stalemate: the exogenous stress or the military and recruitment mechanisms?" Or, more explicitly, "can the military and recruitment mechanisms enhance the unpredictability of the exogenous factors?" The answer to this question is that conflicts can be even more turbulent than their exogenous factors [60–62]. This conclusion is here supported by the study of a particular $D$–$A$ conflict in which the killing time $k_2$ of the second group is assumed to depend on the seasons and therefore vary periodically with a one-year period, i.e.,

$$k_2(t) = \bar{k}_2 \left( 1 + \varepsilon \sin \frac{2\pi}{T} t \right)$$

where $\bar{k}_2$ is the mean value of the killing time, $\varepsilon$ is the seasonal variability, and $T = 52$ if the time unit is the week. Thus, in this particular conflict the environment is not chaotic because the stress $w(t)$ is simply periodic. However, a detailed numerical analysis (see Appendix 5 in S1 File) shows that for suitable parameter values and for sufficiently strong seasonal variability $\varepsilon$ the conflict is chaotic. The result is depicted in Fig 21 where $\varepsilon$ and $\bar{k}_2$ are on the horizontal and vertical axis, respectively. The figure shows that the resulting stalemates can be periodic (green), quasi-periodic (yellow) or chaotic (red). Since on the vertical axis (i.e., when there are no seasons and therefore the killing time is constant) the stalemates are stationary or periodic, the figure shows that seasons trigger aperiodic behaviors. The figure also shows that the degree of seasonality $\varepsilon$ required to generate chaotic stalemates increases with $\bar{k}_2$. A more careful analysis (see Appendix 5 in S1 File) shows that less environmental variability $\varepsilon$ is needed to generate chaos when the period of the periodic stalemate in the absence of seasons is close to 1 year. In conclusion, chaos arises more easily if the military and environmental clocks beat at comparable frequencies, a quite interesting result. Similar results have been found in ecology [63] and interpersonal relationships [64].

## Mixed recruitments

In the previous sections, we have assumed that $A$ groups follow strict attack strategies, i.e., their recruitment is given by Eq (2) with $z_i = L_{i+}$. If we now suppose that such groups are also slightly sensitive to the injuries they suffer, we must use Eq (2) with $z_i = \lambda_i L_{+i} + (1 - \lambda_i) L_{i+}$ with $\lambda_i$ small. Thus, the effect of the propensity to defend can be established by determining how the behavior of a conflict varies when $\lambda_i$ is increased starting from $\lambda_i = 0$, i.e., starting from a pure attack strategy.

The result can be clearly represented by the three-dimensional graph of Fig 22 which shows that the propensity to defend is a stabilizing factor.

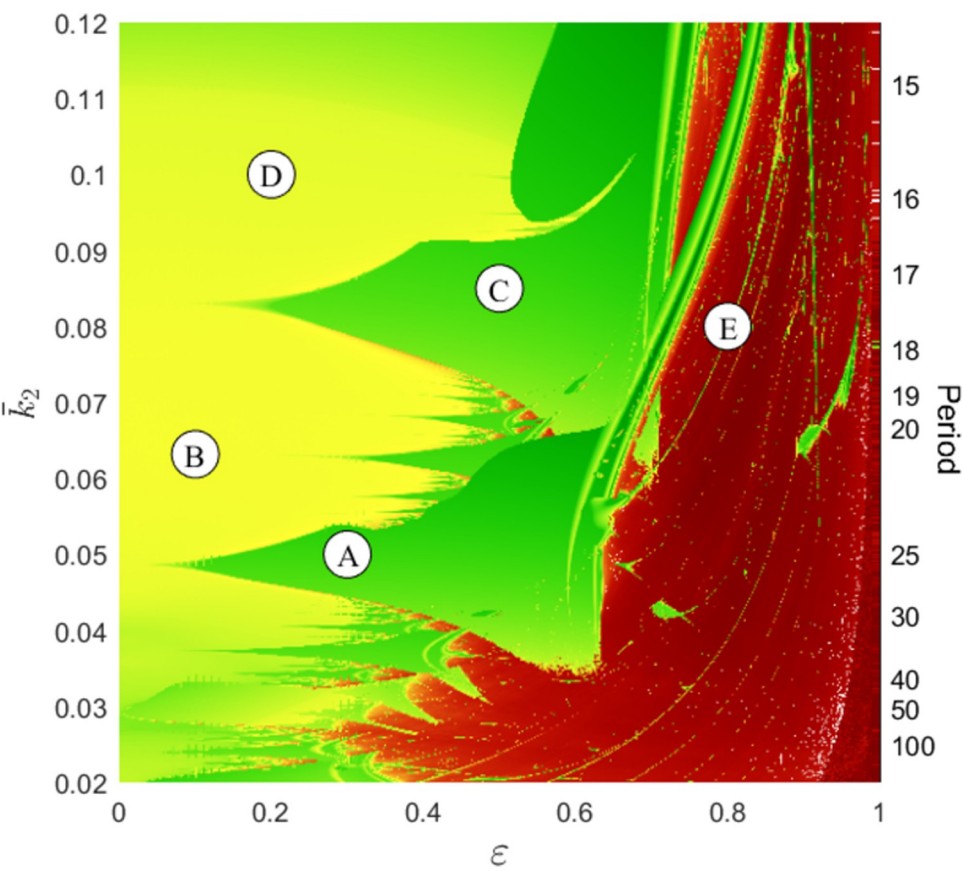

**Fig 21. In the red regions the stalemate is chaotic because the Lyapunov exponent is positive.** In the yellow regions the stalemate is quasi-periodic and in the green regions it is periodic. On the vertical axis (i.e., when there are no seasons) the stalemate is periodic and its period (in weeks) is indicated on the right of the figure. Points A—E correspond to conflicts whose time series are reported in (see Fig A5.1 and Appendix 5 in S1 File).

## Undetectability of the groups

Sometimes *A* groups are not easily detectable because they are hidden in a civil population. In these cases, if $j$ is an *A* group, the loss $l_{ij}$ inflicted to it by one unit of group $i$ is particularly low when $x_j$ is small, because in that case individuals of group *A* are really difficult to be identified. That is to say, when $x_j$ is small, the loss $l_{ij}$ increases less than linearly with $x_j$, i.e., $l_{ij} \sim x_j^{\alpha_j}$ with the undetectability $\alpha_j > 1$. Following the same approach used in Appendix 1 in S1 File to derive Eq (3) we obtain a convex-concave function like that shown in Fig 23 for $\alpha_j = 2$.

Thus, if we like to identify the effect of the undetectability of the individuals of a group, we can produce graphs of the kind shown in Fig 24 for a *D–A* conflict. The figure shows that increasing the undetectability of the *A* group a stationary stalemate is transformed into a periodic stalemate with increasing oscillations, i.e., also undetectability of the group following attack strategies is a destabilizing factor. Thus, the presence of rebel or terrorist groups that hide in civil populations is expected to give rise to very turbulent conflicts. This is a typical feature of guerrilla warfare, where identification of insurgent is problematic [65] and the same insurgent groups stage a limited number of attacks over a prolonged period of time. This maximizes their chance to survive against superior conventional forces and at the same time strains enemy's resources.

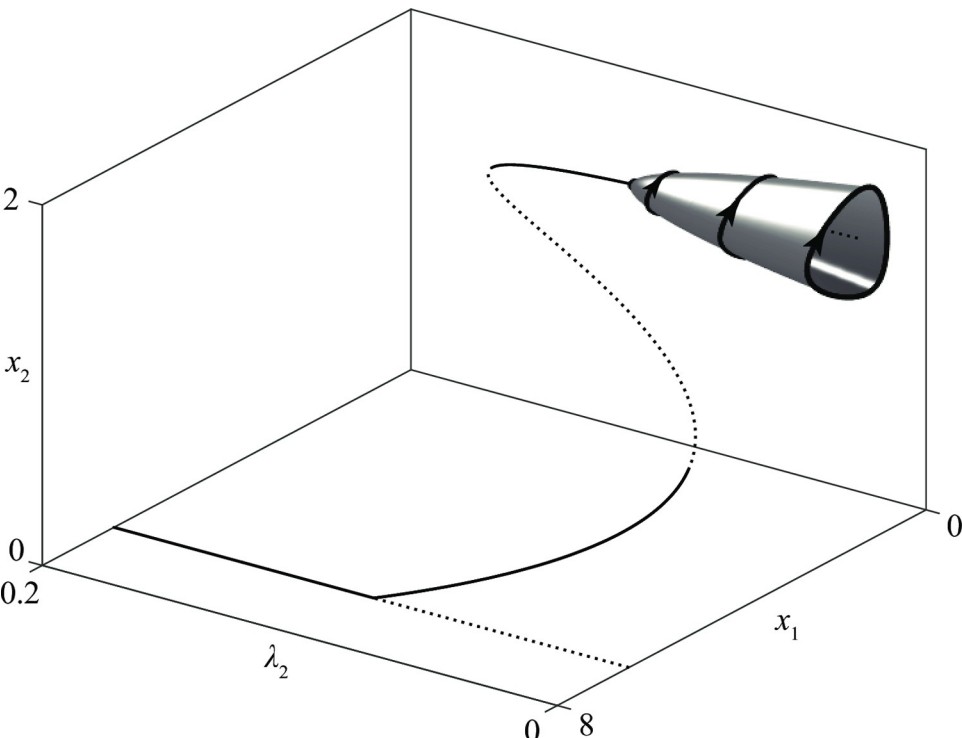

**Fig 22. The case of mixed recruitment strategies.** The influence of propensity to defend of an *A* group in the stalemate of a *D–A* conflict.

### Contagion of attack strategies

The success of a group involved in a conflict can sometimes stimulate other groups involved in other conflicts. Scholarship on terrorism has observed that contagion effects have been at play:

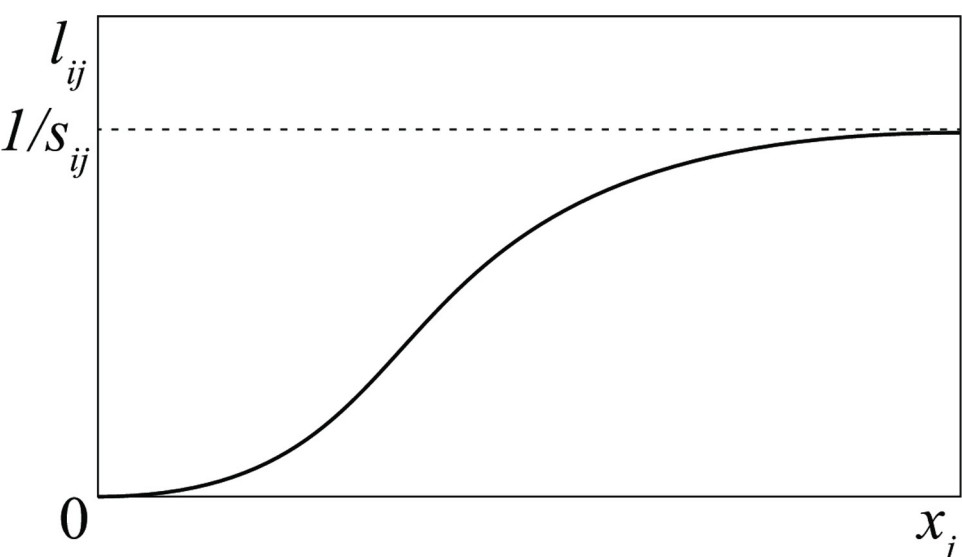

**Fig 23. The loss $l_{ij}$ inflicted to group j by one unit of the group i of its enemies when the group j is not easily detectable because hidden in a civil population.**

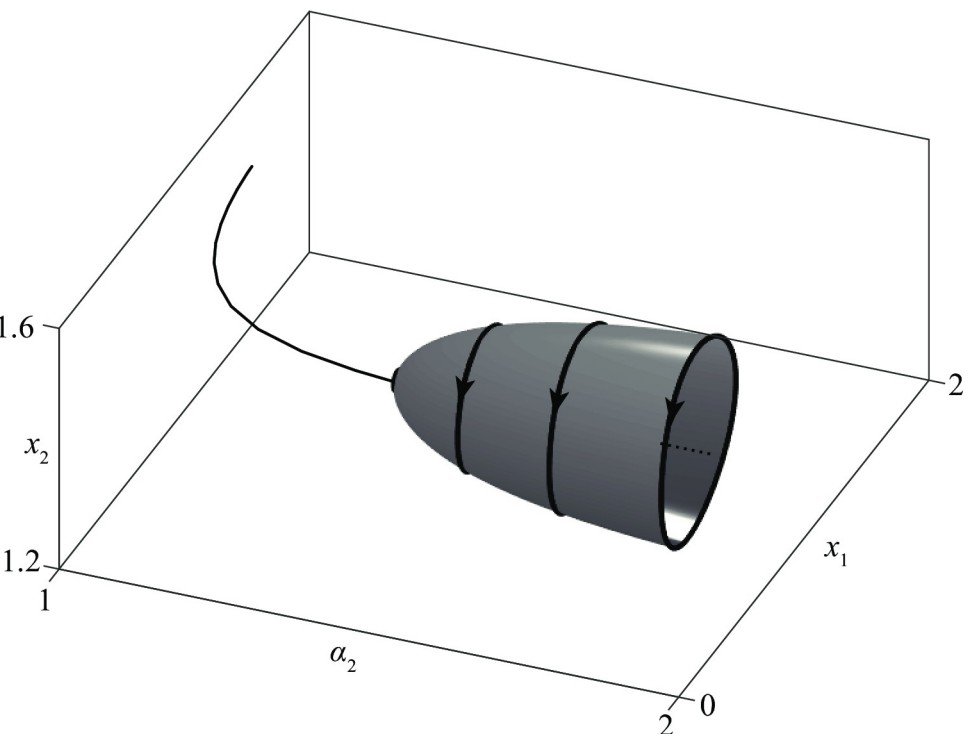

**Fig 24. The influence of undetectability of the *A* group on the stalemate of a *D—A* conflict.**

early studies of terrorism identified that some terrorist techniques were indeed contagious [66] and most recent analyses confirm this finding [67, 68].

The consequences of this sort of spreading of enthusiasm can be theoretically investigated. Consider two independent *D–A* conflicts (see Fig 25A) described by the following four differential equations

$$\dot{x}_1 = b_1 x_1 - c_1 x_1^2 + R_1(L_{21}) - L_{21}$$

$$\dot{x}_2 = b_2 x_2 - c_2 x_2^2 + R_2(L_{21}) - L_{12}$$

$$\dot{x}_3 = b_3 x_3 - c_3 x_3^2 + R_3(L_{43}) - L_{43}$$

$$\dot{x}_4 = b_4 x_4 - c_4 x_4^2 + R_4(L_{43}) - L_{34}$$

and assume that the parameters are such that the two conflicts are trapped in a periodic stalemate (see Fig 25B where the PPP is a single point). Then, assume that the two conflicts are not independent because the attack group of the second conflict (group 4) is positively stimulated by the success of the other attack group. In this case the recruitment $R_4$ depends on a combination (that we assume to be linear) of the numbers of victims of the attack groups, i.e., we replace the recruitment function $R_4(L_{43})$ of the last equation with $R_4(L_{43} + \sigma L_{21})$, where $\sigma$ is a parameter measuring the enthusiasm of group 4 for the success of group 2.

A few simulations of the model, performed for increasing values of $\sigma$, point out that the stalemate of the second conflict, which is periodic for $\sigma = 0$, first becomes quasi-periodic and then chaotic as shown by the PPP's reported in Fig 25C and 25D. This result is in agreement

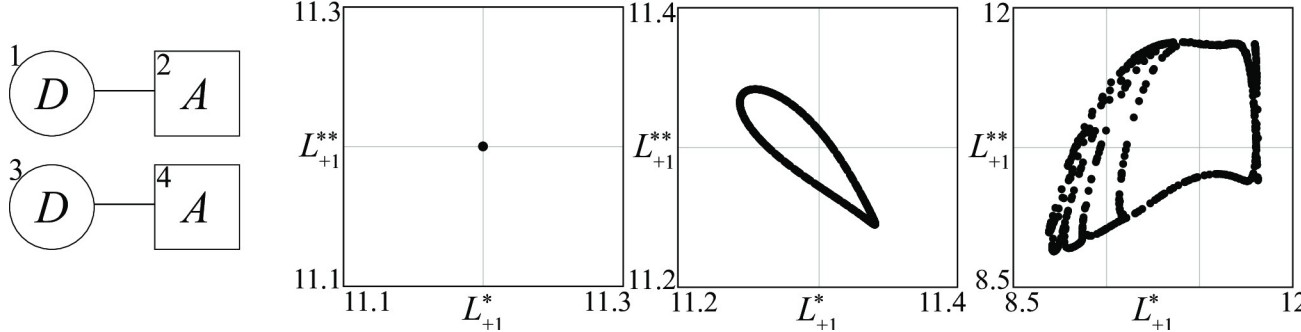

**Fig 25.** Two independent *D—A* conflict (*a*) and three PPP's of the second *D–A* conflict for three increasing values of the spreading of the enthusiasm for attack strategies: (*a*) $\sigma = 0$; (*b*) $\sigma = 0.005$; (*c*) $\sigma = 0.06$.

with our previous analysis of periodically forced *D–A* conflicts (see Fig 21) and shows that also the spreading of enthusiasm for attack strategies is a destabilizing factor.

## Conclusions and extensions

In this paper, we have derived general properties of the time evolution of complex armed conflicts without referring to historical observations, but starting, instead, from simple conjectures on the ideal behaviors of the groups involved in the conflict.

The following are our conjectures. Each group follows a particular recruitment policy and is endowed with specific military characteristics that determine the damages inflicted to the enemies. The recruitment has two components: the basic recruitment which is guaranteed by the "State" even in the absence of enemies and the extra-recruitment that interprets the reaction of the government and of the supporters to the current state of the conflict. The reaction recruitment is bounded but increases with the weighted sum of the suffered and inflicted losses. *D* groups are purely defensive and have weights unbalanced in favor of suffered losses while the opposite is true for *A* groups which follow strictly attack strategies. The losses inflicted by one group to another are also bounded and increase with the sizes of the groups. Representing these conjectures in mathematical terms one obtains an idealized model of the conflict–a set of ordinary differential equations (one for each group). In the idealized model the variables are the sizes of all groups, while the parameters are traits that capture the military and recruitment characteristics of the groups. Thus, studying the idealized model for all parameter settings, is like mimicking all possible conflicts of the world. This is what we have done using dynamical system theory and numerical analysis. For simplicity, we have first studied conflicts between two or three groups and then we have performed a detailed robustness analysis to check that the obtained results continue to hold, at least to a reasonable extent, when the conflict is not as the idealized one.

The following is a partial list of the results we have obtained.

○ Stalemates are possible and can be stationary, periodic, quasi-periodic or chaotic.

○ The fate of the conflict can depend on the initial sizes of the groups.

○ Small but permanent changes of recruitment or military traits can have great consequences and modify the fate of the conflict.

○ Increasing the offensive power of a group can often facilitate its victory (brute force principle).

○ Defensive strategies stabilize conflicts while attack strategies destabilize them.

○ If the attack groups are difficult to detect because hidden in a civil population the conflict becomes more turbulent (terroristic attacks are particularly unpredictable).

○ If a group following attack strategies is sensitive to the success of other $A$ groups (spreading of enthusiasm for attack strategies) the conflict becomes more unpredictable.

○ Conflicts that in the absence of seasonalities are trapped in periodic stalemates with period of the order of the year, can easily become unpredictable in the presence of seasons (seasonalities are destabilizing).

○ A $D$ group trapped in a stalemate can try to get rid of the enemies with a short and heavy attack. If the stalemate is periodic the attack has higher chances of success if it is performed when the losses suffered by the group start raising up.

Even if many of these results are already known because in agreement with empirical observations, the fact that we have derived them from a few general conjectures is particularly interesting. In fact, justifying why something happens can be (in science) more important than knowing that it just happens.

Our idealized model can be used to try to answer a number of open questions. We now mention a few of them that emerge as particularly promising given our results. We start with a couple of questions relevant, at least in principle, to decision makers.

The first question concerns the idea of trying to interrupt a stalemate by supporting the enemy of an enemy. This strategy has been used in many conflicts: for example, during the Yugoslav war, the most lethal conflict was the Bosnian War (1992–1995), a multi-party war that saw Bosnian Muslims pitted against Serbs and Croats. The stalemate was broken when NATO forces engaged in the air power strikes against Serbian troops and militias operating in the conflict, contributing to bring Serbia (formally known as Serbia and Montenegro) to take part in negotiations in US-brokered negotiations held in Dayton, that eventually led to the end of the war [69]. Another case in which governments support militias following the principle of sponsoring the enemy of an enemy are irregular warfare: this is the case of the Colombian government in the fight against the FARC [70]. Our modeling approach can be used to analyze these cases: once a parameter interpreting the support given to the enemy of the enemy is introduced in the model, the reliability of the idea can be checked by performing a bifurcation analysis with respect to that parameter or through a series of simulations.

The second question concerns the possibility of forming coalitions of groups with the aim of having higher chances to eradicate common enemies. For example, in a conflict like the one described in Fig 18A the two $D$ groups could form a temporary coalition to perform a heavy synchronized attack on their common enemy 4. Of course, if the stalemate is chaotic as in Fig 18B, the timing of the attack is important and a suitable extension of the rule discovered for $D$–$A$ conflicts could be the key for the success of the attack. If the common enemy is eradicated, the conflict is split into two independent $D$–$A$ conflicts for which we already know how to proceed with two independent attacks. But we could also consider a second possibility: first eradicate groups 3 and 5 with two independent attacks and only after the success of this operation form the temporary coalition to eradicate the common enemy 4. The idealized model could be used to discover if at least one of the two possibilities is promising.

Other questions are more academic in nature, like the ones concerning the existence of relationships between frequency and severity of military attacks, typically explored with statistical analysis of available data. The most interesting studies along this line, documented in a series of papers by Clauset and coauthors [14, 71, 72], focus on the terrorist attacks in all the

regions of the World during a 40-year-long period. It shows that the data (available in the BAAD data set) obey power law distributions similar to those used to interpret earthquakes and stock-market collapses [12] or follow patterns like the one that can be seen in forest fires [73]. Those power law distributions have been found by analyzing violent events in Iraq (2003–2005), Afghanistan (2008–2010) and Northern Ireland (1969–2001), bringing the scientific community to say that such a distribution can be a universal characteristic of armed conflicts [15, 16] (. A very interesting question that naturally arises at this point is to know if even this result can be the consequence of the simple hypotheses we made on recruitment policies and military characteristics. To answer this question, we should first produce, through simulation, long time series of attacks of a great number of randomly generated independent conflicts, then pack all these series together in a single series and finally analyze it to see if power-law distributions emerge.

Another important research in progress is to better understand and model the effect of spatial dynamics and noise on conflicts. In principle, these effects can be included in our modeling framework, extending our model with a diffusion process on a complex network as typical in meta-population studies [74]. However, since our model has shown sensitive dependence on initial condition, such modeling effort results in a more complex model with low guarantees of a more precise result in terms of model prediction ability.

Instead of looking at the distribution of military attacks at the macroscale, we can focus on a single conflict and ask if some sort of real time forecast can be performed at the microscale. For example: can we predict severity and time of occurrence of the next attack knowing severity and time of occurrence of the last attack? This question seems almost provocative. However, the theory [48] says that in the case of conflicts trapped in chaotic stalemates this could be possible under not very restrictive conditions. We do not enter more deeply into this problem because this would bring us too far. We can only say that in some of the conflicts described in the paper the real time prediction of the attacks is, in principle, possible. Although there are serious doubts on the possibility of transforming these conceptual results into real operational forecasting tools, we believe that further efforts on this issue are worth to be made.

## Supporting information

**S1 File. Appendices 1 to 5. [75].**
(DOCX)

## Author Contributions

**Conceptualization:** Sergio Rinaldi, Alessandra Gragnani, Fabio Della Rossa.

**Data curation:** Alessandra Gragnani, Francesco Niccolò Moro, Fabio Della Rossa.

**Formal analysis:** Sergio Rinaldi, Alessandra Gragnani, Fabio Della Rossa.

**Investigation:** Sergio Rinaldi, Alessandra Gragnani, Francesco Niccolò Moro, Fabio Della Rossa.

**Methodology:** Sergio Rinaldi, Alessandra Gragnani, Fabio Della Rossa.

**Project administration:** Sergio Rinaldi.

**Software:** Alessandra Gragnani, Fabio Della Rossa.

**Supervision:** Sergio Rinaldi, Francesco Niccolò Moro.

**Validation:** Francesco Niccolò Moro.

**Visualization:** Fabio Della Rossa.

**Writing – original draft:** Sergio Rinaldi, Alessandra Gragnani, Francesco Niccolò Moro.

**Writing – review & editing:** Sergio Rinaldi, Alessandra Gragnani, Francesco Niccolò Moro.

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
