## [Decision Letter · Decision Letter 0]

8 Feb 2021

PONE-D-20-37260

A THEORETICAL ANALYSIS OF COMPLEX ARMED CONFLICTS

PLOS ONE

Dear Dr. Della Rossa,

Thank you for submitting your manuscript to PLOS ONE. After careful consideration, we feel that it has merit but does not fully meet PLOS ONE’s publication criteria as it currently stands. Therefore, we invite you to submit a revised version of the manuscript that addresses the points raised during the review process.

Both reviewers indicate some positive points but also both of them remark that the article needs to be carefully rewritten as there are a lot of points to be studied with care. Mainly these problems were with the claims with the relationship to real-world examples.

We look forward to receiving your revised manuscript.

Kind regards,

Roberto Barrio

Academic Editor

PLOS ONE

Journal Requirements:

Reviewers' comments:

Reviewer's Responses to Questions

**Comments to the Author**

1. Is the manuscript technically sound, and do the data support the conclusions?

Reviewer #1: Partly

Reviewer #2: Partly

2. Has the statistical analysis been performed appropriately and rigorously? 

Reviewer #1: N/A

Reviewer #2: N/A

3. Have the authors made all data underlying the findings in their manuscript fully available?

Reviewer #1: No

Reviewer #2: Yes

4. Is the manuscript presented in an intelligible fashion and written in standard English?

Reviewer #1: No

Reviewer #2: Yes

5. Review Comments to the Author

Reviewer #1: Dear Editor,

I have reviewed the manuscript entitled “A theoretical analysis of complex armed conflicts.”

In the paper, the authors propose an idealized dynamical model for conflicts involving multiple group-level actors. They model natural growth, recruitment, and attrition from interaction along with a natural characteristic size to which actors tend in the absence of interaction. They then consider interactions between two types of actors that they distinguish as being either “defensive” or “attack” types in detail for N=2 and N=3. For such analysis, they rely on standard dynamical systems analysis, identifying various fixed points of the system, stable periodic orbits, chaos, and sensitivity to initial conditions. They discuss N>3 as well as various other considerations that might be relevant for armed conflict. They relate aspects of the dynamical system with historical examples of conflict that are evocative of their findings throughout the text.

Overall, I found the model they they wrote down to be interesting with a rich variety of behaviors and that the authors were detailed and methodical in their exploration of the model. This is an interesting extension of the class of models specified by Lanchester. Furthermore, they consider a number of important variation of the model. Finally, the historical allusions were interesting to read in the context of the model (even if they also presented some problems).

As a scientific note, I would ask the authors to address, if briefly, the role of spatial dynamics and noise on conflict dynamics.

Despite these positive comments, there were major issues with the manuscript besides the mathematical model. These problems were with the strong claims made about the relationship to real-world examples, addressing the relevant literature, many grammatical issues, unclear and figures. Though I believe these issues could be fixed, they require substantial manuscript revision. With these shortcomings, I cannot recommend the manuscript for publication at this time.

Please see below for more details.

I would strongly recommend that the authors ask an English-speaking colleague to look over their manuscript. Here, is a non-exhaustive list of some grammatical errors.

In the abstract, “compensating for suffered” and “they inflict onto their enemies.”

Line 35, “defensive” not “defense”

Lines 45-48, run-on sentence

Line 31 non-stationary

There are many such typos that need to be fixed as well as other syntatical issues throughout the text.

Though the historical allusions are nice, the authors are quite strong in their language about the similarity of their model results. For example, lines 221-223 make a general claim about(a) historical conflicts being stalemates and (b) being explained by their model (“This explains why many conflicts…”) The qualitative examples are only an allusion as presented in the text and therefore do not justify such a strong claim. Similar claims are repeated throughout the text. Indeed, the authors emphasized the ideal assumptions made in their model at parts of the paper but were assertive in their claims at other parts, disjunction that should be reconciled.

The authors do not engage with the literature sufficiently. For example, the authors dismiss “traditional descriptive conceptual models,” at the beginning as a contrast to their work. This is an overly general and unspecific claim. Indeed, Richardson did propose a dynamical model (even if it was horribly wrong). Furthermore, such a strong claim warrants much more than two citations—especially if these are “traditional” models then there should be a large literature to cite.

In the discussion, the authors refer to statistical analysis of conflict. Again, there is much literature on statistical analysis of conflict going beyond Clauset and Gleditsch. A sparse sample of examples is

Cederman, L.-E. Modeling the Size of Wars: From Billiard Balls to Sandpiles. APSR 97, 135–150 (2003).

Johnson, N. F. et al. Simple mathematical law benchmarks human confrontations. Sci. Rep. 3, 3463 (2013).

Lee, E. D., Daniels, B. C., Myers, C. R., Krakauer, D. C. & Flack, J. C. Scaling theory of armed-conflict avalanches. Phys. Rev. E 102, 042312 (2020).

Picoli, S., Castillo-Mussot, M. del, Ribeiro, H. V., Lenzi, E. K. & Mendes, R. S. Universal bursty behaviour in human violent conflicts. Sci. Rep. 4, 4773 (2015).

Roberts, D. C. & Turcotte, D. L. Fractality and Self-Organized Criticality of Wars. Fractals 6, 351–357 (1998).

Zammit-Mangion, A., Dewar, M., Kadirkamanathan, V. & Sanguinetti, G. Point process modelling of the Afghan War Diary. Proc. Natl. Acad. Sci. 109, 12414–12419 (2012).

Beyond these examples, the authors do not cite some classic examples of dynamical systems approaches including Lanchester’s work on dynamical models of conflict.

There are classic papers on dynamical systems analysis including chaotic attractors besides Strogatz’s text.

Many of the figures seem unnecessary. Furthermore, some of them are missing labels and legends such Figure 19.

Miscellaneous question:

What happen to the critical points as parameters are changed?

Reviewer #2: This manuscript considers a model for the size of an army, and provides an analysis for a number of cases. It also tries to interpret the dynamical systems analysis using historical examples.

I have two major concerns regarding this text.

First, I have troubles with how the model is presented. Only gradually did I conclude that the army size is main variable. It is stated at lines 80-81, but to understand the introduction a reader needs this earlier. And then finally in the conclusion they present the assumptions as conjectures (lines 631-44). The authors have worked on this topic before, but as a newbie I had a hard time following what it was about. This part should be rewritten so that a math modeller can follow from the start, i.e. assumptions, main variables. It is partially there, but it is scattered. A non-exhaustive list of suggestions:

-l29 Stalemate; would you care to define this? As in population biology, this concerns a attractor with positive state variables I figure?

-l30 "in the sizes and in the losses"; at this point entirely unclear.

-l36 at this point it is unclear what "their recruitment" means. It would not hurt either to explicitly present the 2D ODE system for both D-D and A-D groups. And then show how we see this difference in recruitment. At the end, I still had no idea, and it should be clear from the manuscript, not from work from my side.

-l135,l168; This part on bureaucracy I cannot follow.

-During a stalemate, you would not fight as the fight is fair and you would not gain anything. Yet, the losses in your model are not zero. Please motivate this.

Second, I find the historical reasoning to be too anecdotal. I know modelling and dynamical systems, but am not a history expert. I have checked some statements with an expert on warfare. I would add that at least one such person with historical warfare expertise should be reviewing too. I strongly suggest to make a split in the presentation here. For example, for D-D conflicts, present the bifurcation and its general interpretion, and then with a separate heading discuss historical examples, and then with more argumentation why it fits. Now math and history mingle in a suboptimal manner.

Some considerations

l189 post-Westphalian period, please be more specific. Do you mean the year 1648? But it means more than just that!

l224-227; Caution should be added to this interpretation of the Sparta-Athena war.

l230-1; the use of tanks... Germans used tanks too although late, and had antitank ammunition. You point at a single battle only, so why was that crucial only in 1918, while the first tanks were introduced in September 1916? The Americans joining the battle is irrelevant here? Or the lowered support of the German population?

l233 The Enigma code was crucial for the sea, I agree. But it is a long stretch to reduce this to a perturbation, this would rather be a change in parameter values (higher attack/defense efficiency), i.e. property ii. Move it there, and explain it better.

l290-293; This is not too specific. UK and France started war in 1939, when Poland was invaded and the French had the Maginot-line. The Blitzkrieg through the Benelux to avoid it and get to Dunkurque is not readily captured by your model.

l320 Here too, suddenly in brackets some history is dropped. Now I am from Europe, and after a wrinkle I read on, but what about readers from other parts of the world? Please rewrite it properly, dissecting math from history.

l372-375; This is what we would say from a Western perspective, but I am inclined to say we do not that their willingness decreased. That population did not have much voice nor choice. For the US it was a foreign war, for the VietCong a civil war. It might be wise to expand your description, again.

l454-455; "as observed in many real conflicts." I do not like it that you just state it somewhere in passing. This deserves proper data.

l676-9 I do not follow your arguments on the Yugoslavian war.

Minor issues and typo's:

l27; the following reads nicer "for a review see MacKay (2015))"

l31; are contingent  depend

l31; non-stationary (with hyphen) (line 523 too, I just stopped noting them)

l92; , after large

l115; close to one (unit is awkward)

l156; space before is

l164; that THE parameter k_i

l185; "no details are given"...; be constructive, just say you use this software, and btw, the reference is inadequate as the 2003 version does not support connecting orbit detection. The MCMDS(2008)-paper seems more appropriate.

l187; "are hidden, but avaialable" is not so nice for an open access paper. Just say "are all listed in".

l192-4; "fixed at will": I am lost! I think you fix all other parameters, and vary b_1 and c_2. Or even better, "their influence on the dynamics is explored systematically."

l211 "surrounding"; a bifurcation diagram either includes the phase portraits so that "surrounding" is superfluous OR the caption of Figure 3 is incomplete as it then mentions only one part of the figure.

l211 pointed out  illustrated

l215/6 I wonder whether "are highly sensitive" is adequate. Surely "depend" would fit, but it depends what measure one uses to say it depends a lot.

l256 dimension -> size

l258 I wonder what "characteristic" means in this context.

l268 Westerlands's

l326-7; sentence is not wrong, just hard to read.

l351 encapsulated -> summarized

l363 more rare? If I look at figure 10, I would argue the opposite, the mean loss has a plateau at high values and only temporarily drops.

l382 add both after they

l401 please rephrase, difficult to parse

l419-30; can you squeeze this digression?

l442; other conflict types (instead of types of conflict) for readability.

Figure 14; consider using L_i^k and L_i^{k+1} instead of using * and **

l448 add an before aperiodic

l450 a superscript is missing, either * or k

l464 add "the years" before 2013

l469 delete last "of"

l470 add maximal before Lyapunov

l499 "achieve breakthroughs in stalemates", not interrupt, that would mean stop fighting, always a better idea.

l526 strange dots

l548 Why does your signal w(t) need to follow an ODE? as you want to compute these LE's too? Please specify.

l581 pure (it's an adjective)

l605 analysis do confirm  analyses confirm

l619 Replace the = sign by something more appropriate.

l634 State should be put in " as otherwise it does not apply to guerrilla groups

l634 please clarify "that interprets the overall"

l637 defensive

l642 delete (losses)

l648 "not as crude" seems entirely out of place.

l677 I object to "solved", that is not a neutral term here.

l678 part to  part in?

Appendix 5; I object to the adjective military and environmental for LE's. There are two issues, much as before I suggest to distinguish between math and context. Intrinsic and forced LE's, seem clearer from an analysis point of view.

btw. I looked the ref given in line 839-40, but then it would help if say you're using the "method based on qr-decomposition", if that's what you are using.

l868; clarify "simply replicate"

l935-6; I do not think that periods getting closer has to do with synchrony. The differnce between 1 and 15 weeks is still large. I tried to read it a couple of times, but I get confused as T_cycle=52weeks implies synchrony to me if T_env=1 year.

l965; remove alive

l1109; inconsistent initials with dots

ll141; a journal for an arxiv-preprint ?!?

On the one hand it is an attractive idea to describe just the size of an army over time. On the other hand, the interpretation should be done with much more care. There is much to change in this text, but it might be feasible.

6. PLOS authors have the option to publish the peer review history of their article (what does this mean?). If published, this will include your full peer review and any attached files.

Reviewer #1: No

Reviewer #2: No

---

## [Author Response · Author response to Decision Letter 0]

15 Jun 2021

Dear Prof. Barrio

Please find enclosed the revised version of the paper “A theoretical analysis of complex armed conflicts” that we submit to PlosOne as a research article.

First, we want to acknowledge you for the extension you gave us in order to complete our revision: your understanding in this difficult period of time has been greatly appreciated. 

Then, we want to greatly acknowledge the two Reviewers: as you will see in our rebuttal letter, we agree in practical to all of their criticisms. Their comments are punctual, extremely rich, and helped us to make a revised version of our manuscript in our opinion notably improved with respect to our first submission. If possible, we would like to explicitly acknowledge them in the text of the manuscript, if it will be eventually accepted for publication.

We enjoyed making this revision, and we hope that you and the Reviewers will wind this version of the manuscript suitable for publication in PlosOne.

Best regards

Sergio Rinaldi

Alessandra Gragnani

Fabio Della Rossa

Francesco Moro

Reviewer #1: Dear Editor,

I have reviewed the manuscript entitled “A theoretical analysis of complex armed conflicts.”

In the paper, the authors propose an idealized dynamical model for conflicts involving multiple group-level actors. They model natural growth, recruitment, and attrition from interaction along with a natural characteristic size to which actors tend in the absence of interaction. They then consider interactions between two types of actors that they distinguish as being either “defensive” or “attack” types in detail for N=2 and N=3. For such analysis, they rely on standard dynamical systems analysis, identifying various fixed points of the system, stable periodic orbits, chaos, and sensitivity to initial conditions. They discuss N>3 as well as various other considerations that might be relevant for armed conflict. They relate aspects of the dynamical system with historical examples of conflict that are evocative of their findings throughout the text.

Overall, I found the model they wrote down to be interesting with a rich variety of behaviors and that the authors were detailed and methodical in their exploration of the model. This is an interesting extension of the class of models specified by Lanchester. Furthermore, they consider a number of important variation of the model. Finally, the historical allusions were interesting to read in the context of the model (even if they also presented some problems).

R: We thank the Reviewer for his analysis of our work.

As a scientific note, I would ask the authors to address, if briefly, the role of spatial dynamics and noise on conflict dynamics.

R: Spatial dynamics and noise are not considered in our paper, but they could be in principle included with a small modification of the model. This comment has been included in the revised manuscript. 

Despite these positive comments, there were major issues with the manuscript besides the mathematical model. These problems were with the strong claims made about the relationship to real-world examples, addressing the relevant literature, many grammatical issues, unclear and figures. Though I believe these issues could be fixed, they require substantial manuscript revision. With these shortcomings, I cannot recommend the manuscript for publication at this time.

R: As recognized by the Editor, this is the major general criticism made in the manuscript by the Reviewer. We have addressed this criticism by attenuating many strong claims, by adding new references, and by modifying some figures, as specified below.

Please see below for more details.

I would strongly recommend that the authors ask an English-speaking colleague to look over their manuscript. Here, is a non-exhaustive list of some grammatical errors.

R: We have corrected typographical and grammatical errors as well as ambiguous statements. Moreover, we have asked two English-speaking colleagues to look over the manuscript.

In the abstract, “compensating for suffered” and “they inflict onto their enemies.”

Line 31 non-stationary

Line 35, “defensive” not “defense”

Lines 45-48, run-on sentence

There are many such typos that need to be fixed as well as other syntactical issues throughout the text.

R: We have corrected all the syntactical issues detected by the Reviewer as well as all the others that we have found.

Though the historical allusions are nice, the authors are quite strong in their language about the similarity of their model results. For example, lines 221-223 make a general claim about(a) historical conflicts being stalemates and (b) being explained by their model (“This explains why many conflicts…”) The qualitative examples are only an allusion as presented in the text and therefore do not justify such a strong claim. Similar claims are repeated throughout the text. Indeed, the authors emphasized the ideal assumptions made in their model at parts of the paper but were assertive in their claims at other parts, disjunction that should be reconciled.

R: We agree with the Reviewer that the links between the model and the empirical examples were stated in a too rigid way. We dealt with this criticism in two ways. First, we rephrased the text to avoid excessive claims, showing how cases constitute plausible examples for our results. Second, we thoroughly revised the historical examples. In this way we extended the description to better gauge the links between the results of the model and the examples. Also, we provided references to the literature on conflict studies that has findings compatible with those presented in the text.

The authors do not engage with the literature sufficiently. For example, the authors dismiss “traditional descriptive conceptual models,” at the beginning as a contrast to their work. This is an overly general and unspecific claim. Indeed, Richardson did propose a dynamical model (even if it was horribly wrong). Furthermore, such a strong claim warrants much more than two citations—especially if these are “traditional” models then there should be a large literature to cite.

In the discussion, the authors refer to statistical analysis of conflict. Again, there is much literature on statistical analysis of conflict going beyond Clauset and Gleditsch. A sparse sample of examples is

Cederman, L.-E. Modeling the Size of Wars: From Billiard Balls to Sandpiles. APSR 97, 135–150 (2003).

Johnson, N. F. et al. Simple mathematical law benchmarks human confrontations. Sci. Rep. 3, 3463 (2013).

Lee, E. D., Daniels, B. C., Myers, C. R., Krakauer, D. C. & Flack, J. C. Scaling theory of armed-conflict avalanches. Phys. Rev. E 102, 042312 (2020).

Picoli, S., Castillo-Mussot, M. del, Ribeiro, H. V., Lenzi, E. K. & Mendes, R. S. Universal bursty behaviour in human violent conflicts. Sci. Rep. 4, 4773 (2015).

Roberts, D. C. & Turcotte, D. L. Fractality and Self-Organized Criticality of Wars. Fractals 6, 351–357 (1998).

Zammit-Mangion, A., Dewar, M., Kadirkamanathan, V. & Sanguinetti, G. Point process modelling of the Afghan War Diary. Proc. Natl. Acad. Sci. 109, 12414–12419 (2012).

Beyond these examples, the authors do not cite some classic examples of dynamical systems approaches including Lanchester’s work on dynamical models of conflict.

There are classic papers on dynamical systems analysis including chaotic attractors besides Strogatz’s text.

R: We thank the Reviewer for his comment. Indeed, we can engage the literature much more, even if our manuscript contained already 46 references. Following the Reviewer’s suggestions, we have developed our discussion of the current literature. In particular:

• We have better specified the references of the traditional models at the beginning of our work. However, we did not enter in the details of the review paper by MacKay, explicitly referring it to the reader for a deeper study.

• We have now explicitly cited the main works on classic dynamical modeling of conflicts, namely, the original work by Lanchester (1916) suggested by the Reviewer and the two contributions by Richardson (1919, 1935) that elaborate it.

• We have added most of the suggested references on statistical analysis of conflicts, without entering too much in the details, since we think that this is beyond the scope of our work.

• We have added, where possible, other (classical) references to dynamical systems analysis. When only one reference is provided, we clearly said that this is an example, in order to avoid any misunderstanding that this is the unique reference to the subject. 

The reviewed manuscript now contains 75 references: we hope that the Reviewer finds our analysis satisfactory.

Many of the figures seem unnecessary. Furthermore, some of them are missing labels and legends such Figure 19.

R: We have simplified Figure 19, removing the size of the A group that is unnecessary (second panel in the original Figure). We have added the legend. Labels are reported as the minimum and maximum values of the axis: we preferred to keep this style, since the information in our model are qualitative and not quantitative.

Miscellaneous question:

What happen to the critical points as parameters are changed?

R: Critical points are never mentioned in the paper. If the Reviewer means bifurcation points, a slight change in the parameter generically will result in a small displacement of the bifurcation curves in the parameter space. 

Reviewer #2: This manuscript considers a model for the size of an army, and provides an analysis for a number of cases. It also tries to interpret the dynamical systems analysis using historical examples.

I have two major concerns regarding this text.

First, I have troubles with how the model is presented. Only gradually did I conclude that the army size is main variable. It is stated at lines 80-81, but to understand the introduction a reader needs this earlier. And then finally in the conclusion they present the assumptions as conjectures (lines 631-44). The authors have worked on this topic before, but as a newbie I had a hard time following what it was about. This part should be rewritten so that a math modeller can follow from the start, i.e. assumptions, main variables. It is partially there, but it is scattered.

A non-exhaustive list of suggestions:

-l29 Stalemate; would you care to define this? As in population biology, this concerns a attractor with positive state variables I figure?

-l30 "in the sizes and in the losses"; at this point entirely unclear. 

-l36 at this point it is unclear what "their recruitment" means. It would not hurt either to explicitly present the 2D ODE system for both D-D and A-D groups. And then show how we see this difference in recruitment. At the end, I still had no idea, and it should be clear from the manuscript, not from work from my side.

-l135,l168; This part on bureaucracy I cannot follow.

R: We thank the Reviewer for his comment. However, in the Introduction of the paper we preferred to follow, a descriptive approach, without using the modeling language. Indeed, we have rewritten the first part of the paper keeping the language accessible to non-specialists. For example, we have defined what a stalemate is, not using the correct and technical definition suggested by the Reviewer, but in simple words (conflicts with no winner). Similarly, we have better specified the meaning of “size”, “losses”, “recruitment” and “bureaucracy”, to allow a non-specialized reader to understand the sense of the paper. We have also explained the meaning of the terms “stabilize” and “destabilize” introduced in the abstract.

-During a stalemate, you would not fight as the fight is fair and you would not gain anything. Yet, the losses in your model are not zero. Please motivate this.

R: As the Reviewer correctly grasp, at a stalemate of this kind the sizes of the two groups remain constant because their recruitments are positive and balanced.

Second, I find the historical reasoning to be too anecdotal. I know modelling and dynamical systems, but am not a history expert. I have checked some statements with an expert on warfare. I would add that at least one such person with historical warfare expertise should be reviewing too. I strongly suggest to make a split in the presentation here. For example, for D-D conflicts, present the bifurcation and its general interpretation, and then with a separate heading discuss historical examples, and then with more argumentation why it fits. Now math and history mingle in a suboptimal manner.

R: In recasting empirical examples, we followed the Reviewer’s suggestions to provide more background and better specification of the mechanisms., We understand the Reviewer’s concern about excessive mixing up mathematical modeling and empirical narratives. We tried to strike a balance between this requirement and the aim of connecting the results of the model with historical evidence. Following the Reviewer’s suggestion, in the revised paper we split the presentation, first drawing all the mathematical results and then providing the historical evidence in a separate paragraph. We did not separate sections as this would have made large portions of the texts less accessible to audiences not familiar with technical language, yet we believe the text is now clearer. Below, we detail how we addressed each comment related to specific cases. 

Some considerations

l189 post-Westphalian period, please be more specific. Do you mean the year 1648? But it means more than just that!

R: We removed the reference to post-Westphalian period from this place and referred to potential applications to interstate conflicts grounded on large-N scholarship (Jones, Bremer, and Singer (1996)).

l224-227; Caution should be added to this interpretation of the Sparta-Athena war.

R: We added a word of caution, as suggested. We also clarified the claims context in this section, referring to the role of different types of factors that can be considered as perturbations. Specifically, we picked accidental and intentional factors and provided related examples. 

l230-1; the use of tanks... Germans used tanks too although late, and had antitank ammunition. You point at a single battle only, so why was that crucial only in 1918, while the first tanks were introduced in September 1916? The Americans joining the battle is irrelevant here? Or the lowered support of the German population?

R: We better specified the scope of our claim. As shown below in the same section – and coherently with the model’s claim – the fate of war largely depended on what we label as brute force (as the R. suggests mentioning the role of the Americans). Yet, we also believe “perturbations” has their impact in tried to be more specific on the 1918, changing the empirical reference. The new example refers to the German Spring Offensive in March 1918. Reasons of space prevented us from fully treat the case, yet we believe we were able to explain the key insights about the ability of doctrinal changes –based on existing technologies available to the different sides – to create a breakthrough of the stalemate. 

l233 The Enigma code was crucial for the sea, I agree. But it is a long stretch to reduce this to a perturbation, this would rather be a change in parameter values (higher attack/defense efficiency), i.e. property ii. Move it there, and explain it better.

R: We agree with the Reviewer. We reviewed the historical evidence in view of this comment, and, although the Axes had taken some countermeasures against the cracking of the Enigma code, they never discovered it during the course of the war, so they never restored the ‘original parameter values’ in our model. Following the Reviewer’s comment, we reported it as an example of requirement ii, and explained it better.

l290-293; This is not too specific. UK and France started war in 1939, when Poland was invaded and the French had the Maginot-line. The Blitzkrieg through the Benelux to avoid it and get to Dunkurque is not readily captured by your model.

R: We removed the reference to the specific case and mentioned quantitative literature that shows how democracies have been often unable to extract as many resources as possible from their population due to the need of carefully balancing the war effort with consensus (Reiter and Stam (2002), chapter 5).

l320 Here too, suddenly in brackets some history is dropped. Now I am from Europe, and after a wrinkle I read on, but what about readers from other parts of the world? Please rewrite it properly, dissecting math from history.

R:We developed the sentence about the Russian revolution and more clearly stated the large-N supporting evidence for the claim.

l372-375; This is what we would say from a Western perspective, but I am inclined to say we do not that their willingness decreased. That population did not have much voice nor choice. For the US it was a foreign war, for the VietCong a civil war. It might be wise to expand your description, again.

R: We agree with the Reviewer that the statement was too simplistic, and we revised the text to expand and specify our description. We believe that the Reviewer’s reasoning is coherent with ours. To this end we included a brief treatment of scholarship identifying “motivations” as a factor in wars, borrowing from literature adopting prospect theory and relying on surveys (Nincic (1997)), classic studies blending theory and qualitative analysis (Mack (1975)), and large-N studies focusing specifically on US military campaigns (Hulme and Gartzke (2020)).

We maintained the reference to Vietnam as it represents a particularly interesting case in point, but it should now be clearer that the asymmetry of interests was a key driver of the outcome. This does not entail reducing other factors such as the N. Vietnamese regime to impose its people to participate in the war effort.

l454-455; "as observed in many real conflicts." I do not like it that you just state it somewhere in passing. This deserves proper data.

R: We addressed the Reviewer’s concern in two ways. First, many articles, on internal variation of violence in civil wars, show how its patterns vary across time (and space). Due to reasons of space, we mentioned a well-known review of micro-level scholarship on violence in civil wars (Kalyvas (2008)) that looks at several studies adopting a disaggregated approach to the study of violence in civil wars. Second, at the end of the section we mentioned different studies that look at how a plurality of factors within a conflict might have negative consequences on chances of settlement and duration of peace. 

l676-9 I do not follow your arguments on the Yugoslavian war.

R: We rewrote the paragraph to make it clearer.

Minor issues and typo's:

l27; the following reads nicer "for a review see MacKay (2015))"

l31; are contingent  depend

l31; non-stationary (with hyphen) (line 523 too, I just stopped noting them)

l92; , after large

l115; close to one (unit is awkward)

l156; space before is

l164; that THE parameter k_i

l185; "no details are given"...; be constructive, just say you use this software, and btw, the reference is inadequate as the 2003 version does not support connecting orbit detection. The MCMDS(2008)-paper seems more appropriate.

l187; "are hidden, but available" is not so nice for an open access paper. Just say "are all listed in".

l192-4; "fixed at will": I am lost! I think you fix all other parameters, and vary b_1 and c_2. Or even better, "their influence on the dynamics is explored systematically."

l211 "surrounding"; a bifurcation diagram either includes the phase portraits so that "surrounding" is superfluous OR the caption of Figure 3 is incomplete as it then mentions only one part of the figure.

l211 pointed out  illustrated

l215/6 I wonder whether "are highly sensitive" is adequate. Surely "depend" would fit, but it depends what measure one uses to say it depends a lot.

l256 dimension -> size

l258 I wonder what "characteristic" means in this context.

l268 Westerlands's (last name of the General, it is correct)

l326-7; sentence is not wrong, just hard to read.

l351 encapsulated -> summarized

l363 more rare? If I look at figure 10, I would argue the opposite, the mean loss has a plateau at high values and only temporarily drops. (the subject of the sentence is the oscillations, that become rarer when approaching the homoclinic bifurcation in parameter space)

l382 add both after they

l401 please rephrase, difficult to parse

l419-30; can you squeeze this digression?

l442; other conflict types (instead of types of conflict) for readability.

Figure 14; consider using L_i^k and L_i^{k+1} instead of using * and **

l448 add an before aperiodic

l450 a superscript is missing, either * or k

l464 add "the years" before 2013

l469 delete last "of"

l470 add maximal before Lyapunov

l499 "achieve breakthroughs in stalemates", not interrupt, that would mean stop fighting, always a better idea.

l526 strange dots

l548 Why does your signal w(t) need to follow an ODE? as you want to compute these LE's too? Please specify.

l581 pure (it's an adjective)

l605 analysis do confirm  analyses confirm

l619 Replace the = sign by something more appropriate.

l634 State should be put in " as otherwise it does not apply to guerrilla groups

l634 please clarify "that interprets the overall"

l637 defensive

l642 delete (losses)

l648 "not as crude" seems entirely out of place.

l677 I object to "solved", that is not a neutral term here. 

l678 part to  part in?

Appendix 5; I object to the adjective military and environmental for LE's. There are two issues, much as before I suggest to distinguish between math and context. Intrinsic and forced LE's, seem clearer from an analysis point of view. (the term environmental Lyapunov Exponent has been already used in the literature and we want to present these general arguments in our context.)

btw. I looked the ref given in line 839-40, but then it would help if say you're using the "method based on qr-decomposition", if that's what you are using.

l868; clarify "simply replicate"

l935-6; I do not think that periods getting closer has to do with synchrony. The differnce between 1 and 15 weeks is still large. I tried to read it a couple of times, but I get confused as T_cycle=52weeks implies synchrony to me if T_env=1 year.

l965; remove alive

l1109; inconsistent initials with dots

ll141; a journal for an arxiv-preprint ?!?

R: We greatly acknowledge the Reviewer for her/his punctual observations, that we have accepted for the most part and that have improved the manuscript readability. We have explained in parenthesis next to each line the rare Reviewer’s suggestions that we have not implemented.

On the one hand it is an attractive idea to describe just the size of an army over time. On the other hand, the interpretation should be done with much more care. There is much to change in this text, but it might be feasible.

R: We hope that the Reviewer is satisfied with the revised version of our manuscript.

---

## [Decision Letter · Decision Letter 1]

30 Sep 2021

PONE-D-20-37260R1A THEORETICAL ANALYSIS OF COMPLEX ARMED CONFLICTSPLOS ONE

Dear Dr. Della Rossa,

Thank you for submitting your manuscript to PLOS ONE. After careful consideration, we feel that it has merit but does not fully meet PLOS ONE’s publication criteria as it currently stands. Therefore, we invite you to submit a revised version of the manuscript that addresses the points raised during the review process.

We look forward to receiving your revised manuscript.

Kind regards,

Roberto Barrio

Academic Editor

PLOS ONE

Journal Requirements:

Reviewers' comments:

Reviewer's Responses to Questions

**Comments to the Author**

1. If the authors have adequately addressed your comments raised in a previous round of review and you feel that this manuscript is now acceptable for publication, you may indicate that here to bypass the “Comments to the Author” section, enter your conflict of interest statement in the “Confidential to Editor” section, and submit your "Accept" recommendation.

Reviewer #1: All comments have been addressed

Reviewer #2: All comments have been addressed

2. Is the manuscript technically sound, and do the data support the conclusions?

Reviewer #1: Yes

Reviewer #2: Yes

3. Has the statistical analysis been performed appropriately and rigorously? 

Reviewer #1: Yes

Reviewer #2: N/A

4. Have the authors made all data underlying the findings in their manuscript fully available?

Reviewer #1: Yes

Reviewer #2: Yes

5. Is the manuscript presented in an intelligible fashion and written in standard English?

Reviewer #1: Yes

Reviewer #2: Yes

6. Review Comments to the Author

Reviewer #1: Dear Editor,

I have read the revised manuscript submitted by Rinaldi et al. I appreciate the work the authors have put into revising the manuscript, and I find it much improved.

One of my main concerns previously was that the connection between the mathematical results and the historical examples were too tenuous for the strength of the claims made in the paper. I find that the new presentation is much better and highlights the big picture without overly specific claims.

Given that neither the other reviewer nor I are historical experts, however, it seems important to verify the historical descriptions with another expert.

Another major concern I had was with the dearth of citations of the extensive literature on conflict modeling. My point was in reference to a balance must be struck with citing widely, relevancy and the length of the manuscript. While I still feel that some additional citations are missing, I trust that this is a result of careful and purposeful selection by the authors.

Finally, I found the grammar and language of the revised manuscript much improved. Thank you for proofreading this with an English speaker. There remain a couple sections where the grammar is unusually strange and many typos remain—namely pages 25 and 33-34.

A few remaining questions and comments

Is there a particular reason for choosing the parameters b1 and c2 for Figure 3 instead, for example, of the ratios of these parameters b1/b2 and c1/c2 for the two models?

In lines 29 and 41 “i.e.” is italicized and then not. I believe it does not need to be italicized, but this should at least be consistent.

There are some sentences with extra spaces like in line 201 amongst other places. This should be an easy find and replace correction.

As the other reviewer mentioned, it is important to justify why the size of the army is the important variable.

Line 73 should be “to the Appendix”

Line 371 do the authors mean “size” and not “dimension”?

Figure 19 typo “raising”

Line 569 “etc.” instead of “…”

Line 569 “the” unnecessary before “experience”

Line 572 no “their” before “damages”

Punctuation issues in lines 573-576

Lines 618-619 Is there more explanation of this “interesting result”?

Line 643 and elsewhere, conflicts are described as “wild”. This sounds rather colloquial to this native English speaker. I would suggest a different word.

Line 750, “terroristic attack” -> “terrorist attack”

Reviewer #2: I see that almost all of my comments addressed satisfactorily, I have two issues remaining:

(a) the text relating to operation Barba Rossa, see below

(b) military LE; I have a suggestion below, and leave the decision to the editor.

I read the version highlighting the changes and found a few typos still, the line numbers refer to that version.

p5 l145; bureaucratic bureaucracy

p6 l155 Richardson's models

156-157; models by Lanchester (ref) and Deitchman (reads nicer)

p7 l182 less-> lower

l189 an epidemic (singular), not epidemics (plural)

l218 qualitative  qualitatively

l261 forces about  forces were about

l265-6; use of tanks. When you continue with "such innovations" the flow of your story is interrupted, and hard to read. The tanks-part is isolated now. I cannot decide whether you mean both tanks and Stormtroopers or just the latter.

310 Westmorelands' (it is *his* call, yes, I insist)

l341-6 May I suggest to rewrite this? For instance, the state of the Russian army inially was poor even if they had large numbers, and they had chosen a wrong tactic. The mass killing of officers earlier made warfare coordination for the Russians initially hard too. Reading on Barba Rossa, the brutalness came later, and the mass mobilization came later too when German troops had no ammo nor gas left and supplies were hard to get. You may even wonder why the Germans got that far at all. It appears to me that reality here is so complex that your 5-line description is not adequate.

l367 show  shows

l381 large-N scholarship? I have no clue what you mean. Perhaps you mean large-N studies?

438 objective  objectives

443-446; Seems an unfinished sentence to me.

502 this rules  these rules

607 , ... reads as unfinished

808 have  has

l1215; IBC; with online sources you must state what date you retrieved it.

In the appendix: I still think military LE is a wrong name, and should be avoided. Rather you would like to say this is the intrinsic LE opposed to the environmental LE as it is derived from the "conflict dynamics".

7. PLOS authors have the option to publish the peer review history of their article (what does this mean?). If published, this will include your full peer review and any attached files.

Reviewer #1: No

Reviewer #2: **Yes: **Hil Meijer

---

## [Author Response · Author response to Decision Letter 1]

12 Nov 2021

Dear Prof. Barrio

Please find enclosed the revised version of the paper “A theoretical analysis of complex armed conflicts” that we submit to PlosOne as a research article.

We have taken in consideration all of the Reviewers comments, which we greatly acknowledge for their precious work, that have both increased clarity and readability of our manuscript. We think the paper greatly improved throughout the review process. In the following, you can find the punctual response to each of the Reviewers' comments. We hope you and the Reviewers will find this version of the manuscript suitable for publication in PlosOne.

Best regards

Sergio Rinaldi

Alessandra Gragnani

Fabio Della Rossa

Francesco Moro

Reviewer #1: Dear Editor,

I have read the revised manuscript submitted by Rinaldi et al. I appreciate the work the authors have put into revising the manuscript, and I find it much improved.

One of my main concerns previously was that the connection between the mathematical results and the historical examples were too tenuous for the strength of the claims made in the paper. I find that the new presentation is much better and highlights the big picture without overly specific claims.

Given that neither the other reviewer nor I are historical experts, however, it seems important to verify the historical descriptions with another expert.

Another major concern I had was with the dearth of citations of the extensive literature on conflict modeling. My point was in reference to a balance must be struck with citing widely, relevancy and the length of the manuscript. While I still feel that some additional citations are missing, I trust that this is a result of careful and purposeful selection by the authors.

Finally, I found the grammar and language of the revised manuscript much improved. Thank you for proofreading this with an English speaker. 

We are happy that the Reviewer appreciated the work we did for the revision. Effectively, balance the huge literature on conflict and the length limits of the manuscript was harsh. If the Reviewer has in mind some particular relevant citation we are missing, please tell it to us, so that we can try to add it (or explain why we had not added it).

There remain a couple sections where the grammar is unusually strange and many typos remain—namely pages 25 and 33-34.

We have rephrased page 25 and the last pages of the manuscript.

A few remaining questions and comments

Is there a particular reason for choosing the parameters b1 and c2 for Figure 3 instead, for example, of the ratios of these parameters b1/b2 and c1/c2 for the two models?

Unfortunately, this is not possible, because the ratios between two parameters do not uniquely identify the system dynamics. For example, let c1=5 and c2=2; if b1=b2=1 (b1/b2=1) the system’s dynamics is described by the portrait 5, while if b1=b2=2 (b1/b2=1) the system’s dynamics is described by the portrait 1. 

In lines 29 and 41 “i.e.” is italicized and then not. I believe it does not need to be italicized, but this should at least be consistent.

Following the Reviewer suggestion, we have removed all the italic forms for ‘i.e.’ in the manuscript.

There are some sentences with extra spaces like in line 201 amongst other places. This should be an easy find and replace correction.

Done.

As the other reviewer mentioned, it is important to justify why the size of the army is the important variable.

The size of the group, as stated in the paper, is a suitable variable capturing the group’s war ability. For this reason, we assume that we can use this variable to mathematically describe the conflict dynamics.

Line 73 should be “to the Appendix”

Line 371 do the authors mean “size” and not “dimension”?

Figure 19 typo “raising”

Line 569 “etc.” instead of “…”

Line 569 “the” unnecessary before “experience”

Line 572 no “their” before “damages”

Punctuation issues in lines 573-576

Lines 618-619 Is there more explanation of this “interesting result”?

Line 643 and elsewhere, conflicts are described as “wild”. This sounds rather colloquial to this native English speaker. I would suggest a different word.

Line 750, “terroristic attack” -> “terrorist attack”

Done, thank you.

 

Reviewer #2: I see that almost all of my comments addressed satisfactorily, I have two issues remaining:

(a) the text relating to operation Barba Rossa, see below

(b) military LE; I have a suggestion below, and leave the decision to the editor.

I read the version highlighting the changes and found a few typos still, the line numbers refer to that version.

p5 l145; bureaucratic bureaucracy

p6 l155 Richardson's models

156-157; models by Lanchester (ref) and Deitchman (reads nicer)

p7 l182 less-> lower

l189 an epidemic (singular), not epidemics (plural)

l218 qualitative  qualitatively

l261 forces about  forces were about

310 Westmorelands' (it is *his* call, yes, I insist) (Gotta! Even you misspelled his last name again! Thank you)

l367 show  shows

l381 large-N scholarship? I have no clue what you mean. Perhaps you mean large-N studies?

438 objective  objectives

443-446; Seems an unfinished sentence to me.

502 this rules  these rules

607 , ... reads as unfinished

808 have  has

l1215; IBC; with online sources you must state what date you retrieved it.

We thank the Reviewer for his precise and punctual comments.

l265-6; use of tanks. When you continue with "such innovations" the flow of your story is interrupted, and hard to read. The tanks-part is isolated now. I cannot decide whether you mean both tanks and Stormtroopers or just the latter.

We thank the Reviwer for his comment. Effectively, we were trying to say to many things, and, at the end, we make confusion in the reader. Stromtroopers were the innovation that break stalemate for German, while innovation related to tanks (principally done by British) contribute to break another stalemate. In the revised version of the text, we now only talk about the impact of infantry tactics.

l341-6 May I suggest to rewrite this? For instance, the state of the Russian army initially was poor even if they had large numbers, and they had chosen a wrong tactic. The mass killing of officers earlier made warfare coordination for the Russians initially hard too. Reading on Barba Rossa, the brutalness came later, and the mass mobilization came later too when German troops had no ammo nor gas left and supplies were hard to get. You may even wonder why the Germans got that far at all. It appears to me that reality here is so complex that your 5-line description is not adequate.

We did rewrite and clarify making other factors contributing to early German success explicit (although in respect of word constraints). Yet, we maintained a key element of the previous version. While we don’t deny that other factors contributed to German advance, and it was during the German retreat that most violence effectively occurred, “brutal” management of conquered territories clearly allowed Germans to speed up operations. This did not just occur while retreating (though this was more widespread and visible) but also while advancing. The September-October 1941 Babi Yar massacre – for instance – has been described as part of a broader strategy adopted by German forces to “eliminate pockets of resistance” (the point is made for instance by Shepherd, Ben, War in the Wild East: The German Army and Soviet Partisans, cited in the text but also by Wette, Wolfram, Wehrmacht. History, Myth, Reality, Harvard University Press, 2006 and present in the memoirs of some German officers, Bidermann, Gottlob Herbert. 2000. “In Deadly Combat: A German Soldier’s Memoir of the Eastern Front, Trans.” Derek Zumbro. Lawrence: Univ. Press of Kansas). Statements on Soviet resources mobilized comes from data available in Harrison, M. (1988). Resource mobilization for World War II: the USA, UK, USSR, and Germany, 1938-1945. Economic History Review, 171-192 (reference added). 

In the appendix: I still think military LE is a wrong name, and should be avoided. Rather you would like to say this is the intrinsic LE opposed to the environmental LE as it is derived from the "conflict dynamics".

We do like the term ‘military LE’, that is in accordance also with the terms ‘biological Lyapunov Exponent’ (Colombo et al. 2008) and the terms ‘romantic Lyapunov Exponent’ (Rinaldi et al. 2015).

---

## [Decision Letter · Decision Letter 2]

9 Dec 2021

PONE-D-20-37260R2A THEORETICAL ANALYSIS OF COMPLEX ARMED CONFLICTSPLOS ONE

Dear Dr. Della Rossa,

Thank you for submitting your manuscript to PLOS ONE. After careful consideration, we feel that it has merit but does not fully meet PLOS ONE’s publication criteria as it currently stands. Therefore, we invite you to submit a revised version of the manuscript that addresses the points raised during the review process.

We look forward to receiving your revised manuscript.

Kind regards,

Roberto Barrio

Academic Editor

PLOS ONE

Journal Requirements:

Additional Editor Comments (if provided):

Dear authors

In this version both reviewers like the article.

There are only very few grammatical remarks that have to be addressed.

Once the few modifications have been done the paper will be in a suitable form to be accepted.

Reviewers' comments:

Reviewer's Responses to Questions

**Comments to the Author**

1. If the authors have adequately addressed your comments raised in a previous round of review and you feel that this manuscript is now acceptable for publication, you may indicate that here to bypass the “Comments to the Author” section, enter your conflict of interest statement in the “Confidential to Editor” section, and submit your "Accept" recommendation.

Reviewer #1: All comments have been addressed

Reviewer #2: All comments have been addressed

2. Is the manuscript technically sound, and do the data support the conclusions?

Reviewer #1: Yes

Reviewer #2: Yes

3. Has the statistical analysis been performed appropriately and rigorously? 

Reviewer #1: N/A

Reviewer #2: N/A

4. Have the authors made all data underlying the findings in their manuscript fully available?

Reviewer #1: No

Reviewer #2: Yes

5. Is the manuscript presented in an intelligible fashion and written in standard English?

Reviewer #1: Yes

Reviewer #2: Yes

6. Review Comments to the Author

Reviewer #1: Dear Editor,

I think the manuscript has come a long way, and it is almost ready. I only have a few grammatical comments.

Line numbers refer to the manuscript with the deltas. Brackets indicate my edit.

Lines 151-153: Is there a parenthesis missing? "is unbounded" should be "[are] unbounded"

Line 360: "Many studies confirm[] that"

Figure 21 caption: Is not a more standard way to write "Points A, ..., E" instead "Points A-E"? There is also a period missing at the end.

Line 736: "concerns the idea [of trying] to interrupt"

Line 770: "focus[es] on" since the subject is "The most interesting study"

Sincerely,

Reviewer 1

Reviewer #2: In general, for the dynamical systems part I am satisfied. As for the interpretation, it is at least acceptable from my side. I must note however that the historical interpretations are not something I can really judge.

I suggest the peer reviews should be published such that it is clear that this was not checked by a historian.

I still think intrinsic and extrinsic Lyapunov Exponents is better terminology. Referring to a different paper that something similar was used is not a good argument. Again, I leave this to the editor.

I went through the text and especially the changes. I found one minor issue:

Line 235; looking at Figure 3 panel 2 I observe two saddles, one on the vertical axis, and a "positive" one within the first quadrant. So perhaps be even more careful indicating the "positive" one? Also, the trajectories forming the stable manifold will approach the saddle but never reach it. So I suggest to write "that approach the saddle".

7. PLOS authors have the option to publish the peer review history of their article (what does this mean?). If published, this will include your full peer review and any attached files.

Reviewer #1: No

Reviewer #2: No

---

## [Editor Report · Decision Letter 3]

11 Feb 2022

A THEORETICAL ANALYSIS OF COMPLEX ARMED CONFLICTS

PONE-D-20-37260R3

Dear Dr. Della Rossa,

We’re pleased to inform you that your manuscript has been judged scientifically suitable for publication and will be formally accepted for publication once it meets all outstanding technical requirements.

Kind regards,

Roberto Barrio

Academic Editor

PLOS ONE
---

## [Editor Report · Acceptance letter]

23 Feb 2022

PONE-D-20-37260R3 

A THEORETICAL ANALYSIS OF COMPLEX ARMED CONFLICTS 

Dear Dr. Della Rossa:

I'm pleased to inform you that your manuscript has been deemed suitable for publication in PLOS ONE. Congratulations! Your manuscript is now with our production department. 

Kind regards, 

on behalf of

Dr. Roberto Barrio 

Academic Editor

PLOS ONE